# Recycling Scraps: Improving Private Learning by Leveraging Intermediate Checkpoints

## Abstract

All state-of-the-art (SOTA) differentially private machine learning (DP ML) methods are iterative in nature, and their privacy analyses allow publicly releasing the intermediate training checkpoints. However, DP ML benchmarks, and even practical deployments, typically use only the final training checkpoint to make predictions. In this work, for the first time, we comprehensively explore various methods that aggregate intermediate checkpoints to improve the utility of DP training. Empirically, we demonstrate that checkpoint aggregations provide significant gains in the prediction accuracy over the existing SOTA for CIFAR10 and StackOverflow datasets, and that these gains get magnified in settings with periodically varying training data distributions. For instance, we improve SOTA StackOverflow accuracies to 22.7% (+0.43% absolute) for $\varepsilon = 8.2$, and 23.84% (+0.43%) for $\varepsilon = 18.9$. Theoretically, we show that uniform tail averaging of checkpoints improves the empirical risk minimization bound compared to the last checkpoint of DP-SGD. Lastly, we initiate an exploration into estimating the uncertainty that DP noise adds in the predictions of DP ML models. We prove that, under standard assumptions on the loss function, the sample variance from last few checkpoints provides a good approximation of the variance of the final model of a DP run. Empirically, we show that the last few checkpoints can provide a reasonable lower bound for the variance of a converged DP model.

## 1 Introduction

Machine learning models can unintentionally memorize sensitive information about the data they were trained on, which has led to numerous attacks that extract private information about the training data (Ateniese et al., 2013; Fredrikson et al., 2014; 2015; Carlini et al., 2019; Shejwalkar et al., 2021; Carlini et al., 2021; 2022). For instance, membership inference attacks (Shokri et al., 2017) can infer whether a target sample was used to train a given ML model, while property inference attacks (Melis et al., 2019; Mahloujifar et al., 2022) can infer certain sensitive properties of the training data. To address such privacy risks, literature has introduced various approaches to privacy-preserving ML (Nasr et al., 2018; Shejwalkar & Houmansadr, 2021; Tang et al., 2022). In particular, iterative techniques like differentially private stochastic gradient decent (DP-SGD) (Song et al., 2013; Bassily et al., 2014; Abadi et al., 2016b; McMahan et al., 2017) and DP Follow The Regularized Leader (DP-FTRL) (Kairouz et al., 2021) have become the state-of-the-art for training DP neural networks.

For establishing benchmarks, prior works in DP ML (Abadi et al., 2016b; McMahan et al., 2017; 2018; Thakkar et al., 2019; Erlingsson et al., 2019; Wang et al., 2019b; Zhu & Wang, 2019; Balle et al., 2020; Erlingsson et al., 2020; Papernot et al., 2020; Tramer & Boneh, 2020; Andrew et al., 2021; Kairouz et al., 2021; Amid et al., 2022; De et al., 2022; Feldman et al., 2022) use only the final model output by the DP algorithm. This is also how DP models are deployed in practice (Ramaswamy et al., 2020; McMahan et al., 2022). However, the privacy analyses for the techniques used allow releasing/using all of the intermediate training checkpoints. In this work, we comprehensively study various methods that leverage intermediate checkpoints to 1) improve the utility of DP training, and 2) quantify the uncertainty in DP ML models that is due to the DP noise.

**Accuracy improvement using checkpoints:** We propose two classes of aggregation methods based on aggregating the *parameters* of checkpoints, or their *outputs*. We provide both theoretical and em-

pirical analyses for our aggregation methods. Theoretically, we show that excess empirical risk of the final checkpoint of DP-SGD is $\log(n)$ times more than that of the weighted average of the past $k$ checkpoints. Here, $n$ is the size of dataset. Empirically, we demonstrate significant top-1 accuracy gains due to our aggregations for image classification (CIFAR10) and a next word prediction (StackOverflow) tasks. Specifically, we show that our checkpoints aggregations achieve absolute (relative) prediction accuracy improvements of 3.79% (7.2%) at $\varepsilon = 1$ for CIFAR10 (DP-SGD), and 0.43% (1.9%) at $\varepsilon = 8.2$ for the StackOverflow (DP-FTRLM) SOTA baselines, respectively. We also show that our aggregations significantly reduce the variance in the performance of DP models over training. Finally, we show that these benefits further magnify in more practical settings with periodically varying training data distributions. For instance, we note absolute (relative) accuracy gains of 17.4% (28.6%) at $\varepsilon = 8$ for CIFAR10 over DP-SGD baseline in such a setting.

**Uncertainty quantification using checkpoints:** There are various sources of randomness in a ML training pipeline (Abdar et al., 2021), e.g., choice of initial parameters, dataset, batching, etc. This randomness induces uncertainty in the predictions made using such ML models. In critical domains, e.g., medical diagnosis, self-driving cars and financial market analysis, failing to capture the uncertainty in such predictions can have undesirable repercussions. DP learning adds an additional source of randomness by injecting noise at every training round. Hence, it is paramount to quantify reliability of the DP models, e.g., by quantifying the uncertainty in their predictions.

To this end, we take the first steps towards *quantifying the uncertainty that DP noise adds* to DP ML training. As prior work, Karwa & Vadhan (2017) develop finite sample confidence intervals but for the simpler Gaussian mean estimation problem. Various methods exist for uncertainty quantification in ML-based systems (Mitchell, 1980; Roy et al., 2018; Begoli et al., 2019; Hubschneider et al., 2019; McDermott & Wikle, 2019; Tagasovska & Lopez-Paz, 2019; Wang et al., 2019a; Nair et al., 2020; Ferrando et al., 2022). However, these methods either use specialized (or simpler) model architectures to facilitate uncertainty quantification, or are not directly applicable to quantify the uncertainty in DP ML due to DP noise. For e.g., a common way of uncertainty quantification (Barrientos et al., 2019; Nissim et al., 2007; Brawner & Honaker, 2018; Evans et al., 2020) that we call the *independent runs* method, needs $k$ independent (bootstrap) runs of the ML algorithm. However, repeating a DP ML algorithm multiple times can incur significant privacy and computation costs.

To address the above issue, we propose to use the last $k$ checkpoints of a single run of a DP ML algorithm as a proxy for the $k$ final checkpoints from independent runs. This does not incur any additional privacy cost to the DP ML algorithm. Furthermore, it is useful in practice as it does not incur additional training compute, and can work with any algorithm having intermediate checkpoints.

Theoretically, we consider using the sample variance of a statistic $f(\theta)$ at checkpoints $\theta_{t_1}, \ldots, \theta_{t_k}$ as an estimator of the variance of $f(\theta_{t_k})$, i.e., the statistic at the final checkpoint, and give a bound on the bias of this estimator. As expected, our bound on the bias decreases as the "*burn-in*" time $t_1$ as well as the time between checkpoints both increase. Intuitively, our proof shows that (i) as the burn-in time increases, the marginal distribution of each $\theta_{t_i}$ approaches the distribution of $\theta_{t_k}$, and (ii) as the time between checkpoints increases, any pair $\theta_{t_i}, \theta_{t_j}$ approaches pairwise independence. Both (i) and (ii) are proven via a mixing time bound, which shows that starting from any point distribution $\theta_0$, the Markov chain given by DP-SGD approaches its stationary distribution at a certain rate. Empirically, we show our method provides reasonable lower bounds on the uncertainty quantified using the more accurate (but privacy and computation intensive) method that uses independent runs.

**Related work on Checkpoint aggregations:** (Chen et al., 2017; Izmailov et al., 2018) explore checkpoint aggregation methods to improve performance in (non-DP) ML settings, but observe negligible performance gains. To our knowledge, De et al. (2022) is the only work in the DP ML literature that uses intermediate checkpoints post training. They apply an exponential moving average (EMA) over the checkpoints of DP-SGD, and note non-trivial gains in performance. However, we propose various aggregation methods that outperform EMA on standard benchmarks.

## 2 IMPROVING ACCURACY BY AGGREGATING DP TRAINING CHECKPOINTS

In this section, we describe our checkpoint aggregation methods, followed by the experimental setup we use for evaluation. Next, we detail our experimental results that demonstrate the significant gains in accuracy of DP ML models due to checkpoints aggregations.

**DP Preliminaries:** Differential Privacy (DP) (Dwork et al., 2006) is a notion to quantify the privacy leakage from the outputs of a data analysis procedure. A randomized algorithm $M : \mathcal{D}^* \to \mathcal{Y}$ is $(\varepsilon, \delta)$-DP if, for all neighbouring datasets $D, D' \in \mathcal{D}^*$ (i.e., datasets that differ in one data sample) and all measurable sets of outputs $S \subseteq \mathcal{Y}$, we have $\mathbb{P}\left[M(D) \in S\right] \leq e^{\varepsilon} \cdot \mathbb{P}\left[M(D') \in S\right] + \delta$. We add Gaussian noise to functions of *bounded sensitivity* to ensure DP. We also use *DP's post-processing property*, i.e., any analysis on the outputs of a DP algorithm does not worsen its DP guarantees.

We consider two state-of-the-art DP ML algorithms: 1) *DP-SGD* for the *central learning* setting, i.e., when all data is pooled at a central location, e.g., a server, and 2) *DP-FTRL* for the *federated learning* (FL) setting, i.e., when all private data is distributed across remote collaborative devices. The privacy analyses for both of these techniques involves *composition* across training rounds, allowing the release of all intermediate checkpoints computed during training.

## 2.1 CHECKPOINTS AGGREGATION METHODS

We propose two types of aggregation methods: *parameter aggregation*, and *output aggregation*. Parameter aggregations compute a function of the parameters of intermediate checkpoints from a DP run, and then use the resulting aggregate parameters for inference. On the other hand, output aggregations compute a function of the outputs of the intermediate checkpoints, and use it for making predictions. These classes can encompass a vast majority of possible aggregation algorithms, but for brevity, we experiment with two algorithms in each of the classes: *exponential moving average* and *uniform past_k average* that aggregate parameters, and *predictions average* and *labels majority vote* that aggregate outputs. Note that all of our aggregation algorithms *post-process* the checkpoints, and hence, do not incur any additional privacy cost. Additionally, our aggregation methods are general, i.e., they are applicable to any training algorithm that computes intermediate checkpoints.

### 2.1.1 PARAMETER AGGREGATION METHODS

**Exponential moving average (EMA):** EMA has been previously used (Tan & Le, 2019; Brock et al., 2021) to improve the performance of ML models at inference time. Starting from the last checkpoint of the run, EMA assigns exponentially decaying weights to each of previous checkpoints; the weights are a function of the EMA *coefficient* $\beta_t$ at step $t$. During training, at each step $t$, EMA maintains a moving average $\theta_{ema}^t$ that is a weighted average of $\theta_{ema}^{t-1}$ and the $t^{th}$ checkpoint, $\theta^t$. This is formalized as follows: $\theta_{ema}^t = (1 - \beta_t) \cdot \theta_{ema}^{t-1} + \beta_t \cdot \theta^t$. Following (Tan & Le, 2019; De et al., 2022), we use a warm-up schedule for the EMA coefficient as: $\beta_t = \min\left(\beta, \frac{1+t}{10+t}\right)$.

**Uniform past_k average (UPA):** For step $t$ of training, UPA computes the mean of the past $k$ checkpoints, i.e., checkpoints from steps $[t - (k - 1), t]$. We formalize this as: $\theta_{upa}^{t,k} = \frac{1}{k} \sum_{i=t-(k-1)}^{t} \theta^i$.

### 2.1.2 OUTPUT AGGREGATION METHODS

**Output predictions averaging (OPA):** For a given test sample $\mathbf{x}$, OPA first computes prediction vectors $f_{\theta^i}(\mathbf{x})$ of the past $k$ checkpoints, i.e., checkpoints from steps $\in [t - (k - 1), t]$, averages the prediction vectors, and computes argmax of the average vector as the final output label. We formalize OPA as $\hat{y}_{opa}(\mathbf{x}) = \mathrm{argmax}\left(\frac{1}{k} \sum_{i=t-(k-1)}^{t} f_{\theta^i}(\mathbf{x})\right)$.

**Output labels majority vote (OMV):** For a given test sample $\mathbf{x}$, OMV computes output prediction vectors for $\mathbf{x}$ and the corresponding labels, i.e., argmax $f_{\theta^i}(\mathbf{x})$. Finally, it uses the majority label among the $k$ labels (breaking ties arbitrarily) as its final output label. We formalize OPA as $\hat{y}_{omv}(\mathbf{x}) = \mathrm{Majority}\left(\mathrm{argmax}(f_{\theta^i}(\mathbf{x}))_{i=t-(k-1)}^{t}\right)$.

## 2.2 EXPERIMENTS

### 2.2.1 EXPERIMENTAL SETUP

**Datasets:** We evaluate our checkpoints aggregation algorithms on three benchmark datasets in two different settings: CIFAR10 (Krizhevsky et al., 2009) and CIFAR100 for image classification, and simulated federated learning over StackOverflow (Kaggle, 2018) for next word prediction. Privacy guarantees are sample-level for CIFAR tasks and user-level for StackOverflow (as it is keyed by

users). For experiments with DP, we fix the privacy parameter $\delta$ to $10^{-5}$ on CIFAR-10/CIFAR-100, and $10^{-6}$ on StackOverflow, ensuring that $\delta < n^{-1}$, where $n$ is the number of examples in CIFAR10/CIFAR-100 and the number of users in StackOverflow.

**Model architectures, and training details:** Following the setup of the state-of-the-art (SOTA) in (De et al., 2022), we train a WideResNet-16-4 with depth 16 and width 4 using DP-SGD (Abadi et al., 2016b) in JAXline (Babuschkin et al., 2020) for $\varepsilon \in \{1, 8\}$. For CIFAR100, we follow (De et al., 2022) and *fine-tune the last, classifier layer of* a WideResNet-28-10 pre-trained on ImageNet data. For StackOverflow, we follow SOTA in (Kairouz et al., 2021; Denisov et al., 2022) and train a one-layer LSTM using DP-FTRL in TFF (Abadi et al., 2016a) for $\varepsilon \in \{8.2, 18.9\}$. For UPA, OPA and OMV, we treat $k$, the number of checkpoints to aggregate, as a hyperparameter and tune it using validation data. All our results are computed over 5 runs of each setting. We provide additional details of our experimental setup in Section B.1.

### 2.2.2 EXPERIMENTAL RESULTS

First, we discuss the results for original datasets and then for more practical periodic distribution shifting datasets. Below, the tables present results for the final training round/step, i.e., for the aggregates computed *until* the last step/round, while plots show results over the last $k$ rounds for some $k$ much smaller than total number of training steps/rounds. For StackOverflow, due to large size of its test data, we provide plots for accuracy on validation data and tables with test accuracy.

**CIFAR10 results:** Table 1 and the left-most two plots in Figure 1 present the accuracy gains in CIFAR10 for $\varepsilon \in \{1, 8\}$. For $\varepsilon = 1$, OPA provides the maximum accuracy gain of 3.79%, while for $\varepsilon = 8$, EMA provides maximum gain of 2.23%.[1] We note from Figure 1 that all checkpoints aggregations improve accuracy for all the training steps of DP-SGD for both $\varepsilon$'s. Next, note from Figure 1 that the accuracy of baseline DP-SGD has a high variance across training steps, i.e., based on the hyperparameters, DP-SGD can produce bad/good models which can be undesirable in practice. However, checkpoints aggregations significantly reduce the variance in accuracy of models, and therefore, increase the confidence in the final DP-SGD model.

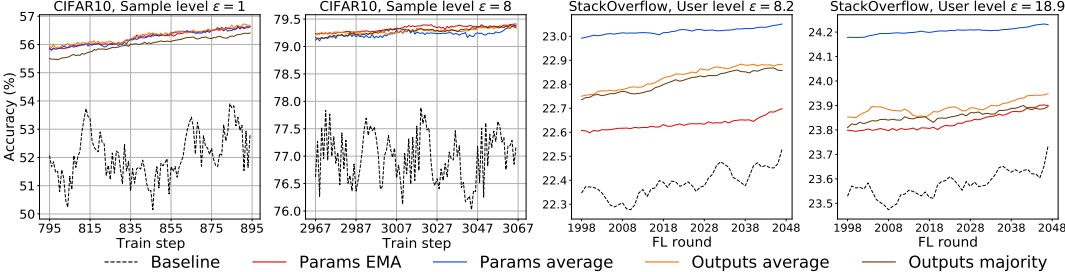

Figure 1: Accuracy improvements due to checkpoints aggregation methods for DP-SGD trained CIFAR10 and DP-FTRL trained StackOverflow.

**StackOverflow results:** Table 1 and the rightmost two plots in Figure 1 present the accuracy gains in StackOverflow for $\varepsilon \in \{18.9, 8.2\}$; Figure 5 present the results for $\varepsilon = \infty$. The maximum accuracy gains are 0.57%, 0.43%, and 0.42% for $\varepsilon$ of $\infty$, 18.9, and 8.2, respectively. We observe that UPA aggregation always provides the best accuracy. Note that these improvements are very significant, because there are 10,004 classes in StackOverflow data. Similarly for CIFAR10, Figures 1 and 5 show that checkpoints aggregations improve accuracy for all training rounds by large margins and also significantly reduce the variance of model accuracy across training rounds. Note that *these improvements are on top of the momentum optimizer* used during training.

**CIFAR100 results:** Due to space constraints, we defer the results to Table 7 in Appendix B, and discuss the major observations here. First, we improve the SOTA baseline of De et al. (2022)[2] from

---

[1]Note that EMA (with default EMA coefficients from non-private settings) has also been used in De et al. (2022) to improve DP-SGD's performance. We observe that even a coarse-grained tuning of the EMA coefficient provides significant accuracy gains. For instance, for CIFAR10, our tuned EMA coefficients outperform those of De et al. (2022) by $\sim 0.6\%$ and 0.35% for $\varepsilon$ of 1 and 8, respectively.

[2]For CIFAR100, Bu et al. (2022) achieve SOTA accuracy of 83% at $\varepsilon = 1$ using 303M parameters vision transformers pre-trained on ImageNet21k. However, due to computational constraints, we use the 36M pa-

Table 1: Test accuracy gains for original CIFAR10 and StackOverflow.

| Privacy level | None (Baseline) | Parameters aggregation | | Outputs aggregation | |
|---|---|---|---|---|---|
| | | EMA | UPA | OPA | OMV |
| CIFAR10; DP-SGD; sample-level privacy | | | | | |
| $\varepsilon = 8$ | $77.18 \pm 1.46$ | $\mathbf{79.41 \pm 0.51}$ | $79.39 \pm 0.52$ | $79.4 \pm 0.59$ | $79.34 \pm 0.54$ |
| $\varepsilon = 1$ | $52.83 \pm 2.17$ | $56.61 \pm 0.91$ | $56.62 \pm 0.89$ | $\mathbf{56.68 \pm 0.89}$ | $56.4 \pm 0.69$ |
| StackOverflow; DP-FTRL; user-level privacy | | | | | |
| $\varepsilon = \infty$ | $25.24 \pm 0.16$ | $25.72 \pm 0.02$ | $\mathbf{25.81 \pm 0.02}$ | $25.79 \pm 0.01$ | $25.78 \pm 0.01$ |
| $\varepsilon = 18.9$ | $23.41 \pm 0.08$ | $23.56 \pm 0.02$ | $\mathbf{23.84 \pm 0.01}$ | $23.6 \pm 0.02$ | $23.57 \pm 0.02$ |
| $\varepsilon = 8.2$ | $22.28 \pm 0.08$ | $22.43 \pm 0.04$ | $\mathbf{22.7 \pm 0.03}$ | $22.57 \pm 0.04$ | $22.52 \pm 0.04$ |

70.6% to 75.51% at $\varepsilon = 1$, and from 77.6% to 80.8% at $\varepsilon = 8$. To achieve these gains, *we perform fine-tuning over the EMA checkpoint of ImageNet-trained WRN-28-10 instead of the final checkpoint* as in (De et al., 2022). Subsequently, we observe that for fine-tuning using CIFAR100, checkpoint aggregations provide small accuracy gains: for both $\varepsilon$ of 1 and 8, we observe maximum gains are 0.11% due to UPA and OPA, respectively.

### 2.2.3 ACCURACY IMPROVEMENTS IN PERIODIC DISTRIBUTION SHIFTING SETTINGS

In many real-world settings, for instance, in federated learning (FL) settings, training data distribution may vary over time. Zhu et al. (2021) demonstrate the adverse impacts of distribution shifts in training data on the performances of resulting FL models. Considering the practical significance of such distribution shifts, we consider settings where the training data distribution has *diurnal* variations, i.e., it is a function of two oscillating distributions, $\mathcal{D}_1$ and $\mathcal{D}_2$; Figure 2 shows these distributions. Such a scenario commonly occurs in FL training, e.g., when a model is trained using FL with client devices participating from two significantly different time zones.

**Experimental setup:** Following Zhu et al. (2021), we consider a setting where training data is a combination of clients/samples drawn from two disjoint data distributions, which oscillate over time as Figure 2 shows. Here the probability of sampling from, e.g., $\mathcal{D}_1$, at time $t$ is $p(\mathcal{D}_1, t) = \left| 2\frac{t \bmod T}{T} - 1 \right|$, where $T$ is the period of oscillation of $\mathcal{D}_{\{1,2\}}$.

To simulate such diurnal distribution, for CIFAR datasets, we design $\mathcal{D}_{\{1,2\}}$ such that $\mathcal{D}_1$ and $\mathcal{D}_2$ respectively contain the data from even and odd classes of the original data. For Stack-Overflow, we design $\mathcal{D}_{\{1,2\}}$ such that $\mathcal{D}_1$ only has the questions from each user, while $\mathcal{D}_2$ only has answers from them.

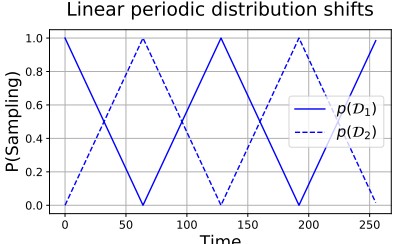

Figure 2: Probability of sampling data from distributions $\mathcal{D}_1$ and $\mathcal{D}_2$.

Then, we draw clients from $\mathcal{D}_{\{1,2\}}$. Apart from data distribution, the rest of experimental setup is the same as before. We use test and validation data same as for the original StackOverflow setting.

**CIFAR10 results:** Table 2 and Figure 3 (left two plots) show accuracy gains for diurnal CIFAR10. We note very large accuracy gains for both $\varepsilon \in \{1, 8\}$; the absolute accuracy gains are 7.45% and 17.37%, respectively, and are due to OPA. Observe that, such diurnal settings have large variances in accuracy across training steps, but our aggregation methods almost completely eliminate the variances. We note that the improvements in diurnal settings are significantly more than that in original CIFAR10. This is because in diurnal settings, the variances in model accuracy over training steps is very large, and hence, the benefits of checkpoints aggregations magnify in these settings.

**StackOverflow results:** Table 2 and Figure 3 (rightmost two plots) present accuracy gains for diurnal StackOverflow. The maximum accuracy gains are 0.09%, 0.44%, and 0.51% for $\varepsilon$ of $\infty$, 18.9, and 8.2, respectively. In contrast to the diurnal CIFAR10 case, improvements in diurnal StackOverflow and StackOverflow are similar. This is because the two distributions in diurnal CIFAR10 (completely different images from even/odd classes) are significantly farther apart compared to the two distributions in diurnal StackOverflow (text from questions/answers).

rameter WRN pre-trained on downsampled ImageNet1k from De et al. (2022). Moreover, similar to De et al. (2022), we observe that in small $\varepsilon$ regimes, fine-tuning only the classifier layer provides better accuracy.

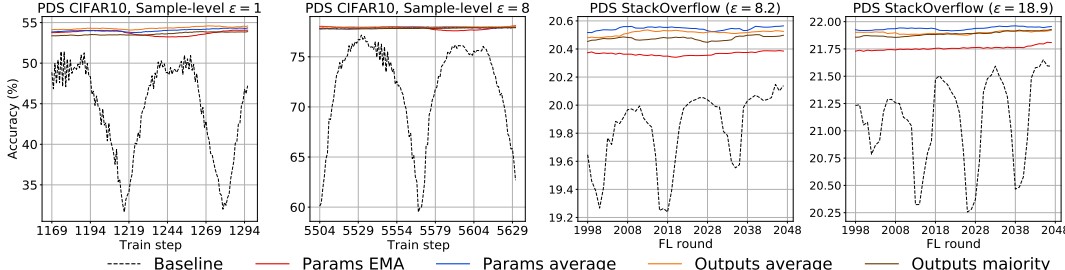

Figure 3: Accuracy gains due to checkpoints aggregations for DP-SGD trained *periodic distribution shifting* (PDS) CIFAR10 (test data) and DP-FTRL trained PDS StackOverflow (validation data).

Table 2: Test accuracy gains for *periodic distribution shifting* (PDS) CIFAR10 and StackOverflow.

| Privacy level | None (Baseline) | Parameters aggregation | | Outputs aggregation | |
|---|---|---|---|---|---|
| | | EMA | UPA | OPA | OMV |
| PDS CIFAR10; DP-SGD; sample-level privacy | | | | | |
| $\varepsilon = 8$ | $60.74 \pm 1.75$ | $78.24 \pm 0.92$ | $77.92 \pm 0.89$ | $\mathbf{78.27 \pm 0.84}$ | $77.99 \pm 0.94$ |
| $\varepsilon = 1$ | $47.13 \pm 1.76$ | $54.04 \pm 0.81$ | $54.35 \pm 0.9$ | $\mathbf{54.58 \pm 0.82}$ | $54.03 \pm 1.08$ |
| PDS StackOverflow; DP-FTRL; user-level privacy | | | | | |
| $\varepsilon = \infty$ | $23.89 \pm 0.14$ | $23.92 \pm 0.12$ | $\mathbf{23.98 \pm 0.02}$ | $23.87 \pm 0.01$ | $23.91 \pm 0.07$ |
| $\varepsilon = 18.9$ | $21.6 \pm 0.13$ | $21.82 \pm 0.07$ | $\mathbf{22.04 \pm 0.11}$ | $21.99 \pm 0.13$ | $21.95 \pm 0.16$ |
| $\varepsilon = 8.2$ | $20.24 \pm 0.29$ | $20.36 \pm 0.06$ | $\mathbf{20.75 \pm 0.05}$ | $20.67 \pm 0.03$ | $20.72 \pm 0.16$ |

**CIFAR100 results:** Due to space constraints, we defer full results in Table 8 in Appendix B. For CIFAR100 also, we observe significant gains due to checkpoints aggregations in PDS setting: for $\varepsilon$ of 1 and 8, we note maximum gains of 4.96% (due to UPA) and 3.37% (due to EMA), respectively. Similar to CIFAR10, the gains due to checkpoints aggregation are significantly higher for PDS-CIFAR100 than for non-PDS CIFAR100.

Table 3: Accuracy gains on test data due to *data-dependent* checkpoints aggregations for Stack-Overflow (SO) and periodic distribution shifting StackOverflow (PDS-SO) trained using DP-FTRL.

| Privacy level | Dataset | Baseline | Best of data-independent | Best of data-dependent |
|---|---|---|---|---|
| $\varepsilon = \infty$ | SO | $25.24 \pm 0.16$ | $25.81 \pm 0.02$ (UPA) | $\mathbf{25.97 \pm 0.01}$ (UPA) |
| | PDS-SO | $23.89 \pm 0.14$ | $23.98 \pm 0.02$ (UPA) | $\mathbf{24.18 \pm 0.02}$ (EMA) |
| $\varepsilon = 18.9$ | SO | $23.41 \pm 0.08$ | $23.84 \pm 0.01$ (UPA) | $\mathbf{23.92 \pm 0.03}$ (UPA) |
| | PDS-SO | $21.60 \pm 0.13$ | $22.04 \pm 0.11$ (UPA) | $\mathbf{22.18 \pm 0.07}$ (EMA) |
| $\varepsilon = 8.2$ | SO | $22.28 \pm 0.08$ | $22.70 \pm 0.03$ (UPA) | $\mathbf{22.80 \pm 0.05}$ (UPA) |
| | PDS-SO | $20.24 \pm 0.29$ | $20.75 \pm 0.05$ (UPA) | $\mathbf{20.90 \pm 0.06}$ (UPA) |

## 2.3 Data-dependent checkpoint selection for aggregation

So far, our aggregated methods operated over a fixed sequence of checkpoints, e.g., past $k$. However, when some public data similar to the private training data is available, we propose a *data-dependent* checkpoint selection method for aggregation: aggregate the checkpoints that perform the best on the (held-out) public data. We expect this to improve over data-independent aggregation schemes. We validate this hypothesis for StackOverflow dataset. We use 10,000 random samples (less than 0.01% of the training data) from the StackOverflow validation data as the held-out public data. Note that this is disjoint from the validation data that we use for hyperparameter tuning. We omit the CIFAR datasets from this evaluation due to the lack of availability of additional similar data.

We compute accuracy of all training checkpoints on the held-out data, find the best $k$ checkpoints, and aggregate them as detailed above. We tune the hyperparameter $k$ using disjoint validation data. Table 3 presents the gains due to data-dependent aggregations over the baseline and data-independent aggregations; Tables 5 and 6 in Appendix B present full results.[3] We see that data-

---

[3]Note that only the data-dependent aggregations use public data, so this comparison is just for illustration.

dependent aggregations outperform data-independent aggregations for all $\varepsilon$'s. For instance, at $\varepsilon$ of 8.2, accuracies of baseline, data-independent and data-dependent aggregations are 22.28%, 22.7% and 22.8%, respectively. For $\varepsilon$'s $\infty$ and 18.9, aggregating best past $k$ checkpoints outperform aggregating past $k$ checkpoints (baseline) by 0.16% (0.73%) and 0.08% (0.51%), respectively. We make the same observation for diurnal PDS StackOverflow dataset—the maximum accuracy gains due to data-dependent aggregations are 0.29%, 0.58%, and 0.66% for $\varepsilon$ of $\infty$, 18.9, and 8.2, respectively.

## 2.4 IMPROVED EXCESS RISK VIA TAIL AVERAGING

In this section, we formally show that checkpoint aggregations like uniform tail averaging provably improves the privacy/utility trade-offs, compared to that of the last checkpoint of DP-SGD. To formalize the problem, we define the following notation: Consider a data set $D = \{d_1, \ldots, d_n\}$ and a loss function $\mathcal{L}(\theta; D) = \frac{1}{n} \sum_{i=1}^{n} \ell(\theta; d_i)$, where each of the loss function $\ell$ is convex and $L$-Lipschitz in the first parameter, and $\theta \in \mathcal{C}$ with $\mathcal{C} \subseteq \mathbb{R}^p$ being a convex constraint set. We analyze the following variant of DP-SGD, which is guaranteed to be $\rho$-zCDP[4].

1. $\theta_0 \leftarrow \mathbf{0}^p$.
2. For $t \in [T]$, $\theta_{t+1} \leftarrow \Pi_{\mathcal{C}} \left( \theta_t - \eta_t \left( \nabla \mathcal{L}(\theta_t; D) + b_t \right) \right)$, where $b_t \sim \mathcal{N} \left( 0, \frac{L^2 T}{2n\rho} \mathbb{I}_{p \times p} \right)$ and $\Pi_{\mathcal{C}} (\cdot)$ being the $\ell_2$-projection onto the set $\mathcal{C}$.

We will provide the utility guarantee for this algorithm by directly appealing to the result of Shamir & Zhang (2013). For a given $\alpha \in (0, 1)$, UPA (Section 2.1.1) corresponds to the average of the last $\alpha T$ models, i.e., $\theta_{\mathsf{upa}}^{\mathsf{priv}} = \frac{1}{\alpha T} \sum_{t=(1-\alpha)T+1}^{T} \theta_t$. One can also consider *polynomial-decay averaging* (PDA) with parameter $\gamma \geq 0$, defined as follows: $\theta_{\mathsf{pda}}^{\mathsf{priv}} [t] = \left( 1 - \frac{\gamma+1}{t+\gamma} \right) \theta_{\mathsf{pda}}^{\mathsf{priv}} [t-1] + \frac{\gamma+1}{t+\gamma} \cdot \theta_t$.

For $\gamma = 0$, PDA matches UPA over all iterates. As $\gamma$ increases, PDA places more weight on later iterates; in particular, if $\gamma = cT$, the averaging scheme is very similar to EMA (Section 2.1.1), since as $t \to T$ the decay parameter $\frac{\gamma+1}{t+\gamma}$ approaches a constant $\frac{c}{c+1}$. In that sense, PDA can be viewed as a method interpolating between UPA and EMA.

**Theorem 2.1** (Adapted from Theorems 2 and 4 of Shamir & Zhang (2013)). *Consider the DP-SGD algorithm above, and the associated parameters. Then there exists choice of learning rate $\eta_t$ and the number of time steps $T$ s.t. the following are true for $\alpha = \Theta(1)$:*

$$\mathbb{E} \left[ \mathcal{L} \left( \theta_{\mathsf{upa}}^{\mathsf{priv}} ; D \right) \right] - \min_{\theta \in \mathcal{C}} \mathcal{L}(\theta; D) = \mathcal{O} \left( \frac{L \|\mathcal{C}\|_2 \sqrt{p}}{n\rho} \right), \text{ and } \mathbb{E} \left[ \mathcal{L}(\theta_T; D) \right] - \min_{\theta \in \mathcal{C}} \mathcal{L}(\theta; D) = \mathcal{O} \left( \frac{L \|\mathcal{C}\|_2 \sqrt{p} \log(n)}{n\rho} \right).$$

*Furthermore, for $\gamma = \Theta(1)$, we have,* $\mathbb{E} \left[ \mathcal{L} \left( \theta_{\mathsf{pda}}^{\mathsf{priv}} [T]; D \right) \right] - \min_{\theta \in \mathcal{C}} \mathcal{L}(\theta; D) = \mathcal{O} \left( \frac{L \|\mathcal{C}\|_2 \sqrt{p}}{n\rho} \right).$

*Proof.* If we choose $T = \lceil n\rho \rceil$ and set $\eta_t$ appropriately, the proof of Theorem 2 (Shamir & Zhang, 2013) implies the following for $\theta_{\mathsf{upa}}^{\mathsf{priv}}$: $\mathbb{E} \left[ \mathcal{L} \left( \theta_{\mathsf{upa}}^{\mathsf{priv}} ; D \right) \right] - \min_{\theta \in \mathcal{C}} \mathcal{L}(\theta; D) = O \left( \frac{L \|\mathcal{C}\|_2 \sqrt{p}}{n\rho} \log \left( \frac{1}{\alpha} \right) \right).$

Setting $\alpha = \Theta(1)$ gives the theorem's first part, and $\alpha T = 1$, i.e., $1/\alpha = T = \lceil n\rho \rceil$ gives the second. The third follows from modifying Theorem 4 of Shamir & Zhang (2013) for the convex case (see the end of Section 4 of Shamir & Zhang (2013) for details). $\qquad\square$

The excess empirical risk for $\theta_T$ is higher by factor of $\log(n)$ in comparison to $\theta_{\mathsf{upa}}^{\mathsf{priv}}$ and $\theta_{\mathsf{pda}}^{\mathsf{priv}} [T]$. For the step size selections typically used in practice (e.g., fixed or inverse polynomial step sizes), the last iterate will suffer from the extra $\log(n)$ factor, and we do not know how to avoid it. Furthermore, Harvey et al. (2019) showed that this is unavoidable in the non-private, high probability regime.[5]

---

[4]Using Bun & Steinke (2016), it is easy to convert the privacy guarantee to an $(\varepsilon, \delta)$-DP guarantee.

[5]Jain et al. (2021) show that for carefully chosen step sizes, the logarithmic factor can be removed, and Feldman et al. (2020) extend this analysis to a DP-SGD variant with varying batch sizes. Unlike those methods, averaging can be done as post-processing of DP-SGD outputs, rather than a modification of the algorithm.

# 3 QUANTIFYING UNCERTAINTY IN PREDICTIONS DUE TO DP NOISE

In this section, we discuss our proposal of how to quantify the uncertainty that the differential privacy (DP) noise adds to the outputs of ML algorithms, without additional privacy cost or computation. First, we discuss the issues with the current established approach of uncertainty quantification in ML when used in DP setting, and then discuss our proposal and theoretical results. Finally, we provide experimental results that demonstrate the practical utility of our approach.

**(Naive) uncertainty quantification, and its hurdles:** As discussed in Section 1, although uncertainty quantification has a long history of research, prior methods, including the most common independent runs method, are not applicable in DP settings. The two major issues with the independent runs method in DP settings are: First, the additional runs of a DP-ML algorithm can incur a significant increase in the privacy cost. Second, the additional computation required in independent runs method can be prohibitive, e.g., in production FL. Hence, it is not feasible to use the naive method for uncertainty quantification in DP settings.

## 3.1 TWO BIRDS, ONE STONE: OUR UNCERTAINTY QUANTIFICATION PROPOSAL

To address the two hurdles discussed above, we propose a simple yet efficient method that leverages intermediate checkpoints computed during a DP run. Specifically, we substitute the $k$ output models from the independent runs method with $k$ checkpoints, e.g., the last $k$ checkpoints, from a single DP run. The rest of the confidence interval computation is the same for both the methods. Specifically, we compute the most likely classification labels of the $k$ models/checkpoints on a test input. Use the set of $k$ output labels as a sample from the *student's $t$-distribution* and compute a confidence interval of the mean of the distribution. Finally, we use the average of multiple such intervals over multiple test inputs as a proxy for uncertainty of outputs of the DP ML algorithm.

### 3.1.1 THEORY

In this section, we give a theoretical motivation for using checkpoints to estimate the variance of a statistic. We show that under certain assumptions, if we take the sample variance of a statistic $f(\theta)$ computed at the checkpoints $\theta_{t_1}, \ldots, \theta_{t_k}$, in expectation it is a good approximation of the sample variance of the limiting distribution of DP-SGD. As expected, the quality of the approximation improves by increasing the length of the "burn-in" time as well as the time between checkpoints. To simplify the proof, we actually prove the theorem for DP-LD, a continuous-time version of DP-SGD. DP-LD can be defined as follows. We first reformulate (unconstrained) DP-SGD with step size $\eta$ as: $\widetilde{\theta}_{(t+1)\eta} \leftarrow \widetilde{\theta}_{t\eta} - \eta \nabla \mathcal{L}(\widetilde{\theta}_{t\eta}; D) + b_t, b_t \sim \mathcal{N}(0, 2\eta\sigma^2 \mathbb{I}_{p \times p})$.

Notice that we have reparameterized $\widetilde{\theta}$ so that its subscript refers to the sum of all step-sizes so far, i.e. after $t$ iterations we have $\widetilde{\theta}_{t\eta}$ and not $\widetilde{\theta}_t$. Also notice that the variance of the noise we added is proportional to the step size $\eta$. In turn, for any $\eta$ that divides $t$, after $t/\eta$ iterations with step size $\eta$, the sum of variances of noises added is $2t\sigma^2$. Now, taking the limit as $\eta$ goes to 0 of the sequence of random variables $\{\widetilde{\theta}_{t\eta}\}_{t \in \mathbb{Z}_{\geq 0}}$ defined by DP-SGD, we get a continuous sequence $\{\theta_t\}_{t \in \mathbb{R}_{\geq 0}}$. In particular, if we fix some $t$, then $\theta_t$ is the limit as $\eta$ goes to 0 of $\widetilde{\theta}_t$ defined by DP-SGD with step size $\eta$. This sequence is exactly the sequence defined by DP-LD, which is more formally given by the following stochastic differential equation:

$$d\theta_t = -\nabla \mathcal{L}(\theta_t; D)dt + \sigma\sqrt{2}dW_t. \tag{1}$$

Here, $W_t$ is a standard Brownian motion and $\sigma^2$ is analogous to the variance of noise added in DP-SGD. In Appendix A we show the following:

**Theorem 3.1** (Simplified version of Theorem A.1)**.** *Suppose $\mathcal{L}$ is 1-strongly convex and $M$-smooth, and $\sigma = 1$ in (1). Let $0 < t_1 < t_2 < \ldots < t_k$ be such that $t_{i+1} \geq t_i + \gamma, \forall i > 0$ and some $\gamma$. Let $\{\theta_{t_i} : i \in [k]\}$ be the checkpoints, and $f : \Theta \to [-1, 1]$ be a statistic whose variance we wish to estimate. Let $S$ be the sample variance of $f$ over the checkpoints, and $V$ be the variance of $f(\theta_{t_k})$, i.e. the statistic at the final checkpoint. Then, for some "burn-in" time $B$ that is a function of $\theta_0, M, p$, we have: $|\mathbb{E}[S] - V| = \exp(-\Omega(\min\{t_1, \gamma\} - B))$.*

Intuitively, Theorem A.1 and its proof say the following: (i) As we increase $t_1$, the time before the first checkpoint, each of the checkpoints' marginal distributions approaches the distribution of $\theta_{t_k}$,

and (ii) As we increase $\gamma$, the time between checkpoints, the checkpoints' distributions approach pairwise independence. So increasing both $t_1$ and $\gamma$ causes our checkpoints to approach $k$ pairwise independent samples from the distribution of $\theta_{t_k}$, i.e., our variance estimator approaches the true variance in expectation. To show both (i) and (ii), we build upon past results from the sampling literature to show a mixing bound of the following form: running DP-LD from any point initialization[6] $\theta_0$, the Rényi divergence between $\theta_t$ and the limit as $t \to \infty$ of DP-LD, $\theta_\infty$, decays exponentially in $t$. This mixing bound shows (i) since if $t_1$ is sufficiently large, then the distributions of all of $\theta_{t_1}, \theta_{t_2}, \ldots, \theta_{t_k}$ are close to $\theta_\infty$, and thus close to each other. This also shows (ii) since DP-LD is a Markov chain, i.e. the distribution of $\theta_{t_j}$ conditioned on $\theta_{t_i}$ is equivalent to the distribution of $\theta_{t_j - t_i}$ if we run DP-LD starting from $\theta_{t_i}$ instead of $\theta_0$. So our mixing bound shows that even after conditioning on $\theta_{t_i}$, $\theta_{t_j}$ has distribution close to $\theta_\infty$. Since $\theta_{t_j}$ is close to $\theta_\infty$ conditioned on any value of $\theta_{t_i}$, then $\theta_{t_j}$ is almost independent of $\theta_{t_i}$.

**Empirical analysis:** We compare the uncertainty quantified using the independent runs method and using our method; experimental setup is the same as in Section 2.2.1. First, for both StackOverflow and CIFAR10 datasets, we do 100 independent training runs. Then to compute uncertainty using the independent runs method, we take the final model from $k$ of these runs (chosen randomly), compute prediction labels for a given input, compute confidence interval widths for the input, and finally, use the average of confidence

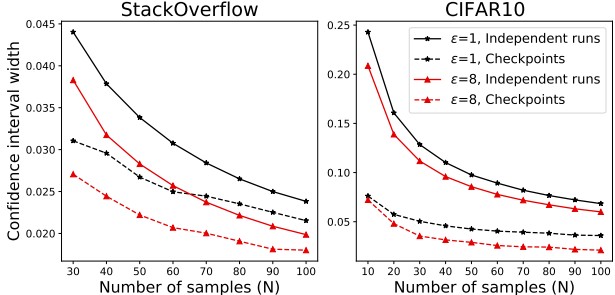

Figure 4: Uncertainty due to DP noise measured using confidence interval widths. We compute the intervals using N bootstrap (independent) runs, and using the last N checkpoints of a single run.

interval widths of a large number of inputs as the final uncertainty estimate. To compute uncertainty using our checkpoints based method, we instead select our $k$ models to be the last $k$ checkpoints of a random training run, and obtain average confidence intervals as before.

Figure 4 shows the results for StackOverflow and CIFAR10. Plots show intervals averaged over 5 runs, i.e., by sampling $k$ final models 5 times or by using the last $k$ models of 5 random runs. We observe that uncertainty computed using the intermediate checkpoints consistently give a reasonable lower bound on the uncertainty computed using the independent runs.

## 4    CONCLUSIONS

In this work, we explore methods for aggregating intermediate checkpoints to improve the utility of DP training. Using checkpoints aggregation methods, we show significant improvements in prediction accuracy over the SOTA for CIFAR and StackOverflow datasets. We also show that uniform tail averaging of checkpoints improves the ERM bound compared to the last checkpoint of DP-SGD. Lastly, we prove that for some standard loss functions, the sample variance from last few checkpoints provides a good approximation of the variance of the final model of a DP run. We also conduct experiments demonstrating that the last few checkpoints can provide a reasonable lower bound for the variance of a converged DP model. For future work, using checkpoint aggregates during DP training could be an interesting direction to further improve its utility. Leveraging intermediate checkpoints to provide variance estimates for checkpoint aggregates could also be a promising direction.

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

## A    PROOF OF THEOREM A.1

For convenience/formality, we review the setup for the theorem we wish to prove. Recall that given a loss function $\mathcal{L}$, by taking the limit as $\eta$ goes to 0 in DP-SGD, we recover the stochastic differential equation (1), restated here for convenience:

$$d\theta_t = -\nabla\mathcal{L}(\theta_t; D)dt + \sigma\sqrt{2}dW_t.$$

Note that the solutions $\theta_t$ to this equation are random variables. A key property of (1) is that the stationary distribution (equivalently, the limiting distribution as $t \to \infty$) has pdf proportional to $\exp(-\mathcal{L}(\theta; D)/\sigma)$ under mild assumptions on $\mathcal{L}$ (which are satisfied by strongly convex and smooth functions).

To simplify proofs and presentation in the section, we will assume that (a) $\theta_0$ is a point distribution, (b) we are looking at unconstrained optimization over $\mathbb{R}^p$, i.e., there is no need for a projection operator in DP-SGD and (1), (c) the loss $\mathcal{L}$ is 1-strongly convex and $M$-smooth, and (d) $\sigma = 1$. We note that (a) can be replaced with $\theta_0$ being sampled from a random initialization without too much work, and (c) can be enforced for Lipschitz, smooth functions by adding a quadratic regularizer. We let $\theta^*$ refer to the (unique) minimizer of $\mathcal{L}$ throughout the section.

Now, we consider the following setup: We obtain a single sample of the trajectory $\{\theta_t : t \in [0, T]\}$. We have some statistic $f : \Theta \to [-1, 1]$, and we wish to estimate the variance of the statistic for the final value $\theta_T$, i.e. the variance $V := \mathbf{Var}(f(\theta_T))$. To do so, we use the sample variance of the checkpoints at times $0 < t_1 < t_2 < t_3 < \ldots < t_k = T$. That is, our estimator is defined as $S = \frac{1}{k-1}\sum_{i=1}^{k}(f(\theta_{t_i}) - \widehat{\mu})^2$ where $\widehat{\mu} = \frac{1}{k}\sum_{i=1}^{k}f(\theta_{t_i})$.

**Theorem A.1.** *Under the preceding assumptions/setup, for some sufficiently large constant $c$, let $\gamma = \frac{1}{2M} + \ln(cM(p + \ln(1/\Delta) + \|\theta_0 - \theta^*\|_2^2)) + c\ln(1/\Delta)$ (recall that $p$ is the dimensionality of the space). Then, if $t_1 > \gamma$ and $t_{i+1} > t_i + \gamma$ for all $i > 0$, for $S, V$ as defined above:*

$$|\mathbb{E}[S] - V| = O(\Delta).$$

Before proving this theorem, we need a few helper lemmas about Rényi divergences:

**Definition A.2.** *The Rényi divergence of order $\alpha > 1$ between two distributions $\mathcal{P}$ and $\mathcal{Q}$ (with support $\mathbb{R}^d$), $D_\alpha(\mathcal{P}, \mathcal{Q})$, is defined as follows:*

$$D_\alpha(P, Q) := \int_{\theta \in \mathbb{R}^d} \frac{P(\theta)^\alpha}{Q(\theta)^{\alpha-1}}d\theta$$

We refer the reader to e.g. van Erven & Harremos (2014); Mironov (2017) for properties of the Rényi divergence. The following property shows that for any two random variables close in Rényi divergence, functions of them are close in expectation:

**Lemma A.3.** *[Adapted from Lemma C.2 of Bun & Steinke (2016)] Let $\mathcal{P}$ and $\mathcal{Q}$ be two distributions on $\Omega$ and $g : \Omega \to [-1, 1]$. Then,*

$$|\mathbb{E}_{x\sim\mathcal{P}}[g(x)] - \mathbb{E}_{x\sim\mathcal{Q}}[g(x)]| \leq \sqrt{e^{D_2(\mathcal{P}||\mathcal{Q})} - 1}.$$

*Here, $D_2(\mathcal{P}||\mathcal{Q})$ corresponds to Rényi divergence of order two between the distributions $\mathcal{P}$ and $\mathcal{Q}$.*

The next lemma shows that the solution to (1) approaches $\theta_\infty$ exponentially quickly in Rényi divergence.

**Lemma A.4.** *Fix some point $\theta_0$. Assume $\mathcal{L}$ is 1-strongly convex, and $M$-smooth. Let $\mathcal{P}$ be the distribution of $\theta_t$ according to (1) for $\sigma = 1$ and:*

$$t := 1/2M + \ln(c(M\|\theta_0 - \theta^*\|_2^2 + p\ln(M))) + c\ln(1/\Delta).$$

*Where $c$ is a sufficiently large constant. Let $\mathcal{Q}$ be the stationary distribution of (1). Then:*

$$D_2(\mathcal{P}, \mathcal{Q}) = O(\Delta^2).$$

The proof of this lemma builds upon techniques in Ganesh & Talwar (2020), and we defer it to the end of the section. Our final helper lemma shows that $\theta_\infty$ is close to $\theta^*$ with high probability:

**Lemma A.5.** *Let $\theta_\infty$ be the random variable given by the stationary distribution of* (1) *for $\sigma = 1$. If $\mathcal{L}$ is 1-strongly convex, then:*

$$\mathbf{Pr}[\|\theta_\infty - \theta^*\|_2 > \sqrt{p} + x] \leq \exp(-x^2/2).$$

*Proof.* We know the stationary distribution has pdf proportional to $\exp(-\mathcal{L}(\theta_t; D))$. In particular, since $\mathcal{L}$ is 1-strongly convex, this means $\theta_\infty$ is a sub-Gaussian random vector (i.e., its dot product with any unit vector is a sub-Gaussian random variable), and thus the above tail bound applies to it. $\qquad\square$

We now will show that under the assumptions in Theorem A.1, every checkpoint is close to the stationary distribution, and that every pair of checkpoints is nearly pairwise independent.

**Lemma A.6.** *Under the assumptions/setup of Theorem A.1, we have:*

*(E1)* $\forall i : |\mathbb{E}[(f(\theta_{t_i}))] - \mathbb{E}[(f(\theta_{t_k}))]| = O(\Delta),$

*(E2)* $\forall i : |\mathbb{E}[(f(\theta_{t_i})^2)] - \mathbb{E}[f(\theta_{t_k})^2]| = O(\Delta),$

*(E3)* $\forall i < j : |\mathbf{Cov}\left(f(\theta_{t_i}), f(\theta_{t_j})\right)| = O(\Delta).$

*Proof.* We assume without loss of generality $\Delta$ is at most a sufficiently small constant; otherwise, since $f$ has range $[-1, 1]$, all of the above quantities can easily be bounded by 2, so a bound of $O(\Delta)$ holds for any distributions on $\{\theta_{t_i}\}$.

For (E1), by triangle inequality, it suffices to prove a bound of $O(\Delta)$ on $|\mathbb{E}[f(\theta_{t_i})] - \mathbb{E}[f(\theta_\infty)]|$. We abuse notation by letting $\theta_t$ denote both the random variable and its distribution. Then:

$$|\mathbb{E}[f(\theta_{t_i})] - \mathbb{E}[f(\theta_\infty)]| \overset{\text{Lemma A.3}}{\leq} \sqrt{e^{D_2(f(\theta_{t_i}), f(\theta_\infty))} - 1} \overset{(*_1)}{\leq} \sqrt{e^{D_2(\theta_{t_i}, \theta_\infty)} - 1}$$
$$\overset{\text{Lemma A.4}}{=} \sqrt{e^{O(\Delta^2)} - 1} \overset{(*_2)}{=} O(\Delta).$$

In $(*_1)$ we use the data-processing inequality (Theorem 9 of van Erven & Harremos (2014)), and in $(*_2)$ we use the fact $e^x - 1 \leq 2x, x \in [0, 1]$ and our assumption on $\Delta$.

(E2) follows from (E1) by just using $f^2$ (which is still bounded in $[-1, 1]$) instead of $f$.

For (E3), note that since (1) is a (continuous) Markov chain, the distribution of $\theta_{t_j}$ conditioned on $\theta_{t_i}$ is the same as the distribution of $\theta_{t_j - t_i}$ according to (1) if we start from $\theta_{t_i}$ instead of $\theta_0$. Let $\mathcal{P}$ be the joint distribution of $\theta_{t_i}, \theta_{t_j}$. Let $\mathcal{Q}$ be the joint distribution of $\theta_{t_i}, \theta_\infty$ (since (1) has the same stationary distribution regardless of its initialization, this is a pair of independent variables). Let $\mathcal{P}', \mathcal{Q}'$ be defined identically to $\mathcal{P}, \mathcal{Q}$, except when sampling $\theta_{t_i}$, if $\|\theta_{t_i} - \theta^*\|_2 > \sqrt{p} + \sqrt{2\ln(1/\Delta)}$ we instead set $\theta_{t_i} = \theta^*$ (and in the case of $\mathcal{P}'$, we instead sample $\theta_{t_j}$ from $\theta_{t_j}|\theta_{t_i} = \theta^*$ when this happens). Let $\mathcal{R}$ denote this distribution over $\theta_{t_i}$. Then similarly to the proof of (E1) we have:

$$|\mathbb{E}_{\mathcal{P}'}[f(\theta_{t_i})f(\theta_{t_j})] - \mathbb{E}_{\mathcal{Q}'}[f(\theta_{t_i})]\mathbb{E}[f(\theta_\infty)]| \overset{\text{Lemma A.3}}{\leq} \sqrt{e^{D_2(\mathcal{P}', \mathcal{Q}')} - 1}$$
$$\overset{(*_3)}{\leq} \sqrt{e^{\max_{\theta_{t_i} \in \text{supp}(\mathcal{R})}\{D_2(\theta_{t_j}|\theta_{t_i}, \theta_\infty)\}} - 1}.$$
$$\overset{\text{Lemma A.4}}{=} \sqrt{e^{O(\Delta^2)} - 1} = O(\Delta).$$

Here $(*_3)$ follows from the convexity of Rényi divergence, and in our application of A.4, we are using the fact that for all $\theta_{t_i} \in \text{supp}(\mathcal{R})$, $\|\theta_{t_i} - \theta^*\|_2 \leq \sqrt{p} + \sqrt{2\ln(1/\Delta)}$. Furthermore, by Lemma A.5, we know $\mathcal{P}$ and $\mathcal{P}'$ (resp. $\mathcal{Q}$ and $\mathcal{Q}'$) differ by at most $\Delta$ in total variation distance. So, since $f$ is bounded in $[-1, 1]$, we have:

$$|\mathbb{E}_{\mathcal{P}}[f(\theta_{t_i})f(\theta_{t_j})] - \mathbb{E}_{\mathcal{P}'}[f(\theta_{t_i})f(\theta_{t_j})]| \leq \Delta,$$

$$|\mathbb{E}_{\mathcal{Q}}[f(\theta_{t_i})]\mathbb{E}[f(\theta_{\infty})] - \mathbb{E}_{\mathcal{Q}'}[f(\theta_{t_i})]\mathbb{E}[f(\theta_{\infty})]| \le \Delta.$$

Then by applying triangle inequality twice:

$$|\mathbb{E}_{\mathcal{P}}[f(\theta_{t_i})f(\theta_{t_j})] - \mathbb{E}_{\mathcal{Q}}[f(\theta_{t_i})]\mathbb{E}[f(\theta_{\infty})]| = O(\Delta)$$

Now we can prove (E3) as follows:

$$
\begin{aligned}
|\mathbf{Cov}\left(f(\theta_{t_i}), f(\theta_{t_j})\right)| &= |\mathbb{E}[(f(\theta_{t_i}) - \mathbb{E}[f(\theta_{t_i})])(f(\theta_{t_j}) - \mathbb{E}[f(\theta_{t_j})])]| \\
&= |\mathbb{E}[f(\theta_{t_i})f(\theta_{t_j})] - \mathbb{E}[f(\theta_{t_i})]\mathbb{E}[f(\theta_{t_j})]| \\
&\le |\mathbb{E}[f(\theta_{t_i})f(\theta_{t_j})] - \mathbb{E}[f(\theta_{t_i})]\mathbb{E}[f(\theta_{\infty})]| + |\mathbb{E}[f(\theta_{t_i})]\mathbb{E}[f(\theta_{\infty})] - \mathbb{E}[f(\theta_{t_i})]\mathbb{E}[f(\theta_{t_j})]| \\
&\le O(\Delta) + |\mathbb{E}[f(\theta_{\infty})] - \mathbb{E}[f(\theta_{t_j})]| = O(\Delta).
\end{aligned}
$$

$\square$

*Proof of Theorem A.1.* We again assume without loss of generality $\Delta$ is at most a sufficiently small constant. The proof strategy will be to express $\mathbb{E}[S]$ in terms of individual variances $\mathbf{Var}\left(f(\theta_{t_i})\right)$, which can be bounded using Lemma A.6.

We have the following:

$$\mathbb{E}[S] = \frac{1}{k-1}\sum_{i=1}^{k}\mathbb{E}\left[(f(\theta_{t_i}) - \widehat{\mu})^2\right] = \frac{1}{k-1}\sum_{i=1}^{k}\mathbb{E}\left[\left(\frac{k-1}{k}\right)^2\left(\underbrace{f(\theta_{t_i})}_{x_i} - \underbrace{\frac{1}{k-1}\sum_{j\in[k],j\neq i}f(\theta_{t_j})}_{y_i}\right)^2\right]. \tag{2}$$

From (2), we have the following:

$$
\begin{aligned}
\mathbb{E}\left[(x_i - y_i)^2\right] &= \mathbb{E}[x_i^2] - 2\mathbb{E}[x_i y_i] + \mathbb{E}[y_i^2] \\
&= \left(\mathbb{E}[x_i^2] - (\mathbb{E}[x_i])^2\right) + \left(\mathbb{E}[y_i^2] - (\mathbb{E}[y_i])^2\right) + \left((\mathbb{E}[x_i])^2 + (\mathbb{E}[y_i])^2 - 2\mathbb{E}\left[x_i y_i\right]\right) \\
&= \underbrace{\mathbf{Var}\left(x_i\right)}_{A} + \underbrace{\mathbf{Var}\left(y_i\right)}_{B} + \underbrace{\left((\mathbb{E}[x_i])^2 + (\mathbb{E}[y_i])^2 - 2\mathbb{E}\left[x_i y_i\right]\right)}_{C}.
\end{aligned}
\tag{3}
$$

In the following, we bound each of the terms $A$, $B$, and $C$ individually. First, let us consider the term $B$. We have the following:

$$B = \mathbf{Var}\left(y_i\right) = \frac{1}{(k-1)^2}\left(\sum_{j\in[k],j\neq i}\mathbf{Var}\left(f(\theta_{t_j})\right) + 2\sum_{\substack{1\le j<\ell\le k \\ j\neq i,\ell\neq i}}\mathbf{Cov}\left(f(\theta_{t_j}), f(\theta_{t_\ell})\right)\right). \tag{4}$$

Plugging Lemma A.6, (E3) into (4) we bound the variance of $y_i$ as follows:

$$B = \mathbf{Var}\left(y_i\right) = \frac{1}{(k-1)^2}\left(\sum_{j\in[k],j\neq i}\mathbf{Var}\left(f(\theta_{t_j})\right)\right) \pm O(\Delta). \tag{5}$$

We now focus on bounding the term $C$ in (3). Lemma A.6, (E1) and (E3) implies the following:

$$(\mathbb{E}[x_i])^2 = (\mathbb{E}[f(\theta_{t_k})])^2 \pm O(\Delta), \tag{6}$$

$$(\mathbb{E}[y_i])^2 = (\mathbb{E}[f(\theta_{t_k})])^2 \pm O(\Delta), \tag{7}$$

$$\mathbb{E}[x_i y_i] = (\mathbb{E}[f(\theta_{t_k})])^2 + O(\Delta). \tag{8}$$

Plugging (6),(7), and (8) into (3), we have

$$\mathbb{E}\left[(x_i - y_i)^2\right] = \mathbf{Var}\left(f(\theta_{t_i})\right) + \frac{1}{(k-1)^2}\left(\sum_{j\in[k],j\neq i}\mathbf{Var}\left(f(\theta_{t_j})\right)\right) \pm O(\Delta). \tag{9}$$

Now, Lemma A.6, (E1) and (E2) implies $\forall i : |\mathbf{Var}\left(f(\theta_{t_i})\right) - \mathbf{Var}\left(f(\theta_{t_k})\right)| = O(\Delta)$. So from (9) we have the following:

$$\mathbb{E}\left[(x_i - y_i)^2\right] = \mathbf{Var}\left(f(\theta_{t_k})\right) \cdot \frac{k}{k-1} \pm O(\Delta). \tag{10}$$

Plugging this bound back in (2), we have the following:

$$\mathbb{E}[S] = \frac{1}{k-1} \cdot \left(\frac{k-1}{k}\right)^2 \cdot k \cdot \left(\mathbf{Var}\left(f(\theta_{t_k})\right) \cdot \frac{k}{k-1} \pm O(\Delta)\right) = \mathbf{Var}\left(f(\theta_{t_k})\right) \pm O(\Delta). \tag{11}$$

Which completes the proof. $\qquad\square$

### A.1    OPTIMIZING THE NUMBER OF CHECKPOINTS

In Theorem A.1, we fixed the number of checkpoints and gave lower bounds on the burn-in time and separation between checkpoints needed for the sample variance bound to have bias at most $\Delta$. We could instead consider the problem where $T$, the time of the final checkpoint, is fixed, and we want to choose $k$ which minimizes the (upper bound on) mean squared error of the sample variance of $\{f(\theta_{iT/k})\}_{i\in[k]}$. Here, we sketch a solution to this problem using the bound from this section.

The mean squared error of the sample variance is the sum of the bias and variance of this estimator. We will use the following reparameterization of Theorem A.1:

**Theorem A.7** (Reparameterized version of Theorem A.1). *Let* $c_1 := \frac{1}{2M} + \ln(c_2 M(p + \|\theta_0 - \theta^*\|_2^2))$, *where* $c_2$ *is a sufficiently large constant. Then if* $S$ *is the sample variance of* $\{f(X_{iT/k})\}_{i\in[k]}$, $V$ *is the true variance of* $X_T$, *and* $T/k > c_1$:

$$|\mathbb{E}[S] - V|^2 \leq \exp\left(-\frac{T/k - c_1}{c_2}\right).$$

One can also show the variance of $S$ is close to the true sample variance:

**Lemma A.8.** *If* $\bar{S}$ *is the sample variance of* $k > 1$ *i.i.d. samples of* $\theta_T$, *then if* $c_2$ *is a sufficiently large constant, for* $c_1$ *as defined in theorem A.7:*

$$\mathbf{Var}\left(\bar{S}\right) \leq \frac{1}{k}, |\mathbf{Var}\left(S\right) - \mathbf{Var}\left(\bar{S}\right)| \leq 2\exp\left(-\frac{T/k - c_1}{c_2}\right).$$

*Proof.* Let $x_1, \ldots, x_k$ be $k$ i.i.d. samples of $f(\theta_T)$, then since each $x_i$ is in the interval $[-1, 1]$:

$$\mathbf{Var}\left(\bar{S}\right) = \frac{\mathbb{E}[x_1^4]}{k} - \frac{\mathbf{Var}\left(x_1\right)(k-3)}{k(k-1)} \leq \frac{1}{k}.$$

Giving the first part of the lemma. For the second part, let $x_i$ be the sampled value of $f(\theta_{iT/k})$. Then:

$$\mathbb{E}[S^2] = \mathbb{E}\left[\left(\left(\frac{1}{k-1}\sum_{i\in[k]}\left(x_i - \frac{1}{k}\sum_{j\in[k]}x_j\right)^2\right)\right)^2\right].$$

For some coefficients $c_{i,j,\ell,m}$, this can be written as $\sum_{i\leq j\leq\ell\leq m}c_{i,j,\ell,m}\mathbb{E}[x_ix_jx_\ell x_m]$ where $\sum_{i\leq j\leq\ell\leq m}|c_{i,j,\ell,m}| \leq 2$. By a similar argument to Theorem A.1, the change in this expectation if we instead use $x_i$ that are i.i.d. is then at most $\exp\left(-\frac{T/k-c_1}{c_2}\right)$ as long as $c_2$ is a sufficiently

large constant. In other words, $|\mathbb{E}[S^2] - \mathbb{E}[\bar{S}^2]| \leq \exp\left(-\frac{T/k - c_1}{c_2}\right)$. A similar argument applies to $E[S]^2$, giving the second part of the lemma. $\qquad\square$

Putting it all together, we have an upper bound on the mean squared error of the sample variance of:

$$\frac{1}{k} + 3\exp\left(-\frac{T/k - c_1}{c_2}\right),$$

Assuming $k > 1, T/k > c_1$. Minimizing this expression with respect to $k$ gives

$$k = \frac{T}{c_1 + c_2 \ln(3T/c_2)},$$

which we can then round to the nearest integer larger than 1 to determine the number of checkpoints to use that minimizes our upper bound on the mean squared error. Of course, if $T < 2c_1$ then Theorem A.1 cannot be applied to give a meaningful bias bound for any number of checkpoints, so this choice of $k$ is not meaningful in that case.

## A.2 PROOF OF LEMMA A.4

We will bound the divergences $D_\alpha(P_1, P_2), D_\alpha(P_2, P_3), D_\alpha(P_3, P_4)$ where $P_1$ is the distribution $\theta_\eta$ that is the solution to (1), $P_2$ is a Gaussian centered at the point $\theta_0 - \eta \nabla \mathcal{L}(\theta_0; D)$, $P_3$ is a Gaussian centered at $\theta^*$, and $P_4$ is the stationary distribution of (1). Then, we can use the approximate triangle inequality for Rényi divergences to convert these pairwise bounds into the desired bound.

**Lemma A.9.** *Fix some $\theta_0$. Let $P_1$ be the distribution of $\theta_\eta$ that is the solution to (1), and let $P_2$ be the distribution $N(\theta_0 - \eta \nabla \mathcal{L}(\theta_0; D), 2\eta)$. Then:*

$$D_\alpha(P_1, P_2) = O\left(M^2 \ln(\alpha) \cdot \max\{p\eta^2, \|\theta_0 - \theta^*\|_2^2 \eta^3\}\right)$$

*Proof.* Let $\theta_t$ be the solution trajectory of (1) starting from $\theta_0$, and let $\theta_t'$ be the solution trajectory if we replace $\nabla \mathcal{L}(\theta_t; D)$ with $\nabla \mathcal{L}(\theta_0; D)$. Then $\theta_\eta$ is distributed according to $P_1$ and $\theta_\eta'$ is distributed according to $P_2$.

By a tail bound on Brownian motion (see e.g. Fact 32 in Ganesh & Talwar (2020)), we have that $\max_{t \in [0,\eta]} \left\|\int_0^t dW_s ds\right\|_2 \leq \sqrt{\eta(p + 2\ln(2/\delta))}$ w.p. $1 - \delta$. Then following the proof of Lemma 13 in Ganesh & Talwar (2020), w.p. $1 - \delta$,

$$\max_{t \in [0,\eta]} \|\theta_t - \theta_0\|_2 \leq cM(\sqrt{p} + \sqrt{\ln(1/\delta)})\sqrt{\eta} + M \|\theta_0 - \theta^*\|_2 \eta,$$

for some sufficiently large constant $c$, and the same is true w.p. $1 - \delta$ over $\theta_t'$. Now, following the proof of Theorem 15 in Ganesh & Talwar (2020), for some constant $c'$, we have the divergence bound $D_\alpha(P_1, P_2) \leq \varepsilon$ as long as:

$$\frac{M^4 \ln^2 \alpha}{\varepsilon^2}(p\eta^2 + \|\theta_0 - \theta^*\|_2^2 \eta^3) < c'.$$

In other words, for any fixed $\eta$, we get a divergence bound of:

$$D_\alpha(P_1, P_2) = O\left(M^2 \ln(\alpha) \cdot \max\{p\eta^2, \|\theta_0 - \theta^*\|_2^2 \eta^3\}\right),$$

as desired. $\qquad\square$

**Lemma A.10.** *Let $P_2$ be the distribution $N(\theta_0 - \eta \nabla \mathcal{L}(\theta_0; D), 2\eta)$ and $P_3$ be the distribution $N(\theta^*, 2\eta)$. Then for $\eta \leq 2/M$:*

$$D_\alpha(P_2, P_3) \leq \frac{\alpha \|\theta_0 - \theta^*\|_2^2}{4\eta}.$$

*Proof.* By contractivity of gradient descent we have:

$$\|\theta_0 - \eta\nabla\mathcal{L}(\theta_0; D) - \theta^*\|_2 \leq \|\theta - \theta^*\|_2.$$

Now the lemma follows from Rényi divergence bounds between Gaussians (see e.g., Example 3 of van Erven & Harremos (2014)). □

**Lemma A.11.** *Let $P_3$ be the distribution $N(\theta^*, 2\eta)$ and let $P_4$ be the stationary distribution of* (1). *Then for $\eta \leq 1/2M$ we have:*

$$D_\alpha(P_3, P_4) \leq \frac{\alpha}{\alpha - 1}\left(\frac{p}{2}\ln(1/\eta) - \ln(2\pi)\right) + \frac{p}{2}\ln(\alpha/4\pi\eta).$$

*Proof.* We have $P_3(\theta) = P_3(\theta^*)\exp(-\frac{1}{4\eta}\|\theta - \theta^*\|_2^2)$ where $P_3(\theta^*) = \left(\frac{1}{4\pi\eta}\right)^d$. By $M$-smoothness of the negative log density of $P_4$, we also have $P_4(\theta) \geq P_4(\theta^*)\exp(-\frac{M}{2}\|\theta - \theta^*\|_2^2)$. In addition, since $P_4$ is 1-strongly log concave, $P_4(\theta^*) \geq \left(\frac{1}{2\pi}\right)^{p/2}$ (as the 1-strongly log concave density with mode $\theta^*$ that minimizes $P_4(\theta^*)$ is the multivariate normal with mean $\theta^*$ and identity covariance). Finally, for $\alpha \geq 1$ and $\eta \leq 1/2M$, we have $\alpha/4\eta > (\alpha - 1)M/2$. Putting it all together:

$$\begin{aligned}
\exp((\alpha - 1)D_\alpha(P_3, P_4)) &= \int \frac{P_3(\theta)^\alpha}{P_4(\theta)^{\alpha-1}}d\theta \\
&= \frac{P_3(\theta^*)^\alpha}{P_4(\theta^*)^{\alpha-1}}\int \exp\left(-(\frac{\alpha}{4\eta} - (\alpha-1)\frac{M}{2})\|\theta - \theta^*\|_2^2\right)d\theta \\
&\leq \left(\frac{1}{4\pi\eta}\right)^{\alpha p/2}(2\pi)^{\alpha(p-1)/2}\int \exp\left(-(\frac{\alpha}{4\eta} - (\alpha-1)\frac{M}{2})\|\theta - \theta^*\|_2^2\right)d\theta \\
&= \left(\frac{1}{2\pi}\right)^{\alpha/2}\left(\frac{1}{2\eta}\right)^{\alpha p/2}\int \exp\left(-(\frac{\alpha}{4\eta} - (\alpha-1)\frac{M}{2})\|\theta - \theta^*\|_2^2\right)d\theta \\
&\stackrel{(*)}{=} \left(\frac{1}{2\pi}\right)^{\alpha/2}\left(\frac{1}{2\eta}\right)^{\alpha p/2}\left(\frac{\frac{\alpha}{4\eta} - (\alpha-1)\frac{M}{2}}{\pi}\right)^{p/2} \\
&\leq \left(\frac{1}{2\pi}\right)^{\alpha/2}\left(\frac{1}{2\eta}\right)^{\alpha p/2}\left(\frac{\alpha}{4\pi\eta}\right)^{p/2} \\
\implies D_\alpha(P_3, P_4) &\leq \frac{\alpha}{\alpha-1}\left(\frac{p}{2}\ln(1/\eta) - \ln(2\pi)\right) + \frac{p}{2}\ln(\alpha/4\pi\eta).
\end{aligned}$$

In $(*)$, we use the fact that $\alpha/4\eta > (\alpha - 1)M/2$ to ensure the integral converges.

□

**Lemma A.12.** *Fix some point $\theta_0$. Let $P$ be the distribution $\theta_\eta$ that is the solution to* (1) *from $\theta_0$ for time $\eta \leq 1/2M$. Let $Q$ be the stationary distribution of* (1). *Then:*

$$D_\alpha(P, Q) = O\left(M^2\ln(\alpha)\cdot\max\{p\eta^2, \|\theta_0 - \theta^*\|_2^2\eta^3\} + \frac{\alpha\|\theta_0 - \theta^*\|_2^2}{\eta} + p\ln(\alpha/\eta).\right)$$

*Proof.* By monotonicity of Rényi divergences (see e.g., Proposition 9 of Mironov (2017)), we can assume $\alpha \geq 2$. Then by applying twice the approximate triangle inequality for Rényi divergences (see e.g. Proposition 11 of Mironov (2017)), we get:

$$D_\alpha(P_1, P_4) \leq \frac{5}{3}D_{3\alpha}(P_1, P_2) + \frac{4}{3}D_{3\alpha-1}(P_2, P_3) + D_{3\alpha-2}(P_3, P_4).$$

The lemma now follows by Lemmas A.9, A.10, A.11. □

Lemma A.4 now follows by plugging $\alpha = 2, \eta = 1/2M$ into Lemma A.12 and then using Theorem 2 of Vempala & Wibisono (2019).

Table 4: StackOverflow LSTM architecture details.

| Layer | Output shape | Parameters |
|---|---|---|
| Input | 20 | 0 |
| Embedding | (20, 96) | 960384 |
| LSTM | (20, 670) | 2055560 |
| Dense | (20, 96) | 64416 |
| Dense | (20, 10004) | 970388 |
| Softmax | - | - |

## B   MISSING DETAILS FROM SECTION 2.2

Below we provide the experimental setup and results that are missing from Section 2.2.

### B.1   EXPERIMENTAL SETUP

**Training hyperparameters:**

*CIFAR10 training*: We use Jaxline (Bradbury et al., 2018) to train on CIFAR10 using DP-SGD (Berrada et al., 2022). For CIFAR10, we use a WideResNet with depth 16 and width 4. We fix clip norm to 1, batch size to 4096 and augmentation multiplicity to 16 as in (De et al., 2022). Then, we set learning rate and noise multiplier, respectively, to 2 and 10 for $\varepsilon = 1$ and to 4 and 3 for $\varepsilon = 8$. For periodic distribution shifting (PDS) CIFAR10, we set learning rate and noise multiplier, respectively, to 2 and 12 for $\varepsilon = 1$ and to 4 and 4 for $\varepsilon = 8$. We stop the training when the intended privacy budget exhausts.

*CIFAR100 training*: For CIFAR100 also, we use Jaxline (Bradbury et al., 2018) and use DP-SGD to *fine-tune the last, classifier layer of* a WideResNet with depth 28 and width 10 that is pre-trained on entire ImageNet data. We fix clip norm to 1, batch size to 16,384 and augmentation multiplicity to 16 as in (De et al., 2022). Then, we set learning rate and noise multiplier, respectively, to 3.5 and 21.1 for $\varepsilon = 1$ and to 4 and 9.4 for $\varepsilon = 8$. For periodic distribution shifting (PDS) CIFAR100, we set learning rate and noise multiplier, respectively, to 4 and 21.1 for $\varepsilon = 1$ and to 5 and 9.4 for $\varepsilon = 8$. We stop the training when privacy budget exhausts.

We would like to highlight that we obtain a significant improvement over the SOTA baseline of De et al. (2022): In particular, *unlike in (De et al., 2022), we fine-tune the final EMA checkpoint*, i.e., the one computed using EMA during pre-training over ImageNet. *This modification (without any additional checkpoints aggregations) gives a major accuracy boost of 5% (70.3% → 75.51%) for $\varepsilon = 1$ and of 3.2% (77.6% → 80.81%) for $\varepsilon = 1$ for the normal CIFAR100 baseline*. We obtain similarly high improvements by fine-tuning the EMA of pre-trained checkpoints (instead of just the final checkpoint) for the PDS-CIFAR100 case. We leave the further investigation of this phenomena to the future work.

*StackOverflow training*: We follow the SOTA in (Kairouz et al., 2021; Denisov et al., 2022) and use TFF to train a one-layer LSTM (detailed architecture in Table 4 (Reddi et al., 2020)) on Stack-Overflow using DP-FTRLM full version from (Denisov et al., 2022). StackOverflow is naturally user-partitioned data, and we process 100 clients in each FL round. We train for 2048 FL rounds and set clip norm, noise multiplier, server learning rate, client learning rate, and server momentum, respectively, to 1, 0.341, 0.5, 1.0, 0.95 for $\varepsilon = 18.9$ and to 1, 0.682, 0.25, 1.0, 0.95 for $\varepsilon = 8.2$. For PDS-StackOverflow, the same set of hyperparameter performs the best based on our tuning of the aforementioned hyperparameters.

**Hyperparameters tuning of checkpoints aggregations:** Here we provide the methodology we follow to obtain the best hyperparameters for our checkpoints aggregation methods (Section 2.1). For EMA, De et al. (2022) simply use the EMA coefficient that works the best in non-private baseline. However, for each of the settings we consider, we tune EMA coefficient in {0.85, 0.9, 0.95, 0.99, 0.999, 0.9999} and observe that the best EMA coefficients for private and non-private settings need not be the same (Table 9). For instance, for CIFAR10, for $\varepsilon$ of 1 and 8, EMA coefficient of 0.95 and 0.99 perform the best and outperform 0.9999 by 0.6% and 0.3%, respectively. Hence, we advise future works to perform tuning of EMA coefficient. Full results are given in Table 9.

Table 5: Accuracy gains on test data due to *data-dependent* checkpoints aggregation algorithms for StackOverflow trained using DPFTRLM to achieve user-level DP.

| Privacy level | Aggregation algorithms | | | | |
|---|---|---|---|---|---|
| | None | Parameters aggregation | | Outputs aggregation | |
| | (Baseline) | EMA | UPA | OPA | OMV |
| $\varepsilon = \infty$ | $25.24 \pm 0.16$ | $25.97 \pm 0.01$ | $\mathbf{25.97 \pm 0.01}$ | $25.94 \pm 0.03$ | $25.97 \pm 0.02$ |
| $\varepsilon = 18.9$ | $23.41 \pm 0.08$ | $23.88 \pm 0.04$ | $\mathbf{23.92 \pm 0.03}$ | $23.81 \pm 0.06$ | $23.76 \pm 0.07$ |
| $\varepsilon = 8.2$ | $22.28 \pm 0.08$ | $22.66 \pm 0.05$ | $\mathbf{22.80 \pm 0.05}$ | $22.7 \pm 0.01$ | $22.69 \pm 0.01$ |

Table 6: Accuracy gains on test data due to *data-dependent* checkpoints aggregation algorithms for *periodic distribution shifting* StackOverflow trained using DPFTRLM to achieve user-level DP.

| Privacy level | Aggregation algorithms | | | | |
|---|---|---|---|---|---|
| | None | Parameters aggregation | | Outputs aggregation | |
| | (Baseline) | EMA | UPA | OPA | OMV |
| $\varepsilon = \infty$ | $23.89 \pm 0.04$ | $\mathbf{24.18 \pm 0.02}$ | $24.15 \pm 0.02$ | $24.15 \pm 0.01$ | $24.19 \pm 0.01$ |
| $\varepsilon = 18.9$ | $21.6 \pm 0.13$ | $\mathbf{22.18 \pm 0.07}$ | $22.17 \pm 0.06$ | $22.21 \pm 0.08$ | $22.19 \pm 0.09$ |
| $\varepsilon = 8.2$ | $20.24 \pm 0.29$ | $20.84 \pm 0.13$ | $20.85 \pm 0.12$ | $\mathbf{20.90 \pm 0.06}$ | $20.82 \pm 0.08$ |

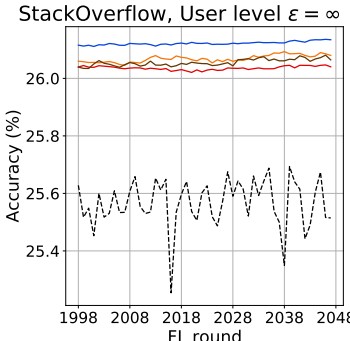 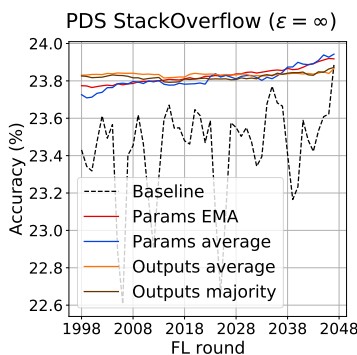

Figure 5: Accuracy gains due to checkpoints aggregations for StackOverflow (left) and periodic distribution shifting StackOverflow (right) trained using DP-FTRLM *without* any DP, i.e., $\varepsilon = \infty$.

For the past $k$ checkpoints aggregation based methods, our tuning methodology is general and as follows: we use some validation data (which is disjoint from training data) and evaluate the efficacy of aggregating $k$ checkpoints where we vary $k$ over a large enough range, and select the $k$ that performs the best on average over 5 runs. Finally, we present results on test data using the best $k$ value. Our hyperparameters tuning method is easy to replicate hence for conciseness, here we only provide the ranges of $k$ that we use for tuning. For CIFAR10 and CIFAR100, we tune $k \in \{3, 5, 10, 20, ..., 200\}$ for both parameters and outputs aggregation. For StackOverflow, we tune $k \in \{3, 5, 10, 20, ..., 200\}$ for parameters aggregations (i.e., UPA) and for outputs aggregation (i.e., OPA and OMV). However, as for outputs aggregations, one should store $k$ checkpoints on device in the FL setting of StackOverflow, we reduce tuning range and tune $k \in \{3, 5, 10, 20, ..., 100\}$.

**Details of data dependent checkpoints selection for StackOverflow:** Here we provide precise method for selecting the best $k$ of all checkpoints and aggregate them. We perform these experiments for StackOverflow due to availability of additional data, and omit CIFAR datasets. Specifically, we use 10,000 samples from the validation partition of the original StackOverflow data and use it as *held-out* data. Note that held-out data is disjoint from both training and validation data we use for other parts of experiments. We compute performance of all checkpoints on held-out data and order them from highest to lowest performing checkpoints. Then we tune $k$ as detailed in the previous section. The aggregation function remains the same for all, but EMA, our aggregation methods from Section 2.1.

Recall that traditional EMA is computed over all the checkpoints computed during training. We compute EMA over best $k$ checkpoints as $\theta_{ema}^i = (1 - \beta) \cdot \theta_{ema}^{i-1} + \beta \cdot \theta^i$, where $i \in \{1, 2, ..., k\}$. We keep EMA coefficient, $\beta$, constant through out. For EMA, we further tune $\beta$ using validation data.

Table 7: Accuracy gains on test data for *CIFAR100 fine-tuned* using DP-SGD and *sample-level* DP.

| Privacy level | None (Baseline) | Parameters aggregation | | Outputs aggregation | |
|---|---|---|---|---|---|
| | | EMA | UPA | OPA | OMV |
| $\varepsilon = 8$ | $80.81 \pm 0.11$ | $80.88 \pm 0.10$ | $80.83 \pm 0.09$ | $\mathbf{80.92 \pm 0.10}$ | $80.82 \pm 0.10$ |
| $\varepsilon = 1$ | $75.51 \pm 0.15$ | $75.42 \pm 0.13$ | $\mathbf{75.62 \pm 0.12}$ | $75.51 \pm 0.16$ | $75.57 \pm 0.18$ |

Table 8: Accuracy gains on test data for *periodic distribution shifting CIFAR100 fine-tuned* using DP-SGD and *sample-level* DP.

| Privacy level | None (Baseline) | Parameters aggregation | | Outputs aggregation | |
|---|---|---|---|---|---|
| | | EMA | UPA | OPA | OMV |
| $\varepsilon = 8$ | $77.16 \pm 0.11$ | $\mathbf{80.53 \pm 0.07}$ | $80.53 \pm 0.08$ | $80.49 \pm 0.06$ | $80.41 \pm 0.09$ |
| $\varepsilon = 1$ | $70.84 \pm 0.16$ | $74.83 \pm 0.15$ | $\mathbf{75.81 \pm 0.16}$ | $75.02 \pm 0.17$ | $74.97 \pm 0.18$ |

Table 9: We observe that tuning the EMA coefficient can provide significant gains in accuracy over the default value of 0.9999 that De et al. (2022) use; with warm-up schedule and number of training steps that De et al. (2022) use, 0.9999 and 0.999 provide the same results. This implies that tuning EMA coefficients for each different privacy budget is required for the best performances.

| Privacy level | EMA coefficient | | | |
|---|---|---|---|---|
| | 0.9 | 0.95 | 0.99 | 0.999 (De et al. (2022)) |
| $\varepsilon = 8$ | 79.41 | 79.35 | **79.41** | 79.16 |
| $\varepsilon = 1$ | 56.59 | **56.61** | 56.06 | 56.05 |

