# OpenReview forum: "Recycling Scraps: Improving Private Learning by Leveraging Intermediate Checkpoints"
_ICLR.cc/2023/Conference — Submitted to ICLR 2023_

### Official Review · Reviewer_374t · 2022-10-22

**Confidence:** 3
**Clarity, Quality, Novelty And Reproducibility:** The paper is well-written, the idea s…
**Correctness:** 4
**Technical Novelty And Significance:** 2
**Empirical Novelty And Significance:** 2
**Recommendation:** 5

**Strength And Weaknesses:**

Strength. Overall, I think this is a solid paper with consistent and extensive experiments. It also provides some theory explaining the insight of their method.

Weakness. (1) The aggregation of checkpoint seems to be fairly straightforward idea of post-processing. (2) The experiment improvement seems marginal. (3) The theory only explains the insight at the convex optimization domain, as I mentioned, there are a tons of related literature and I believe they have better rate in convex optimization setting.

**Summary Of The Paper:**

The paper studies the problem of differential private machine learning (DP ML) and propose to aggreagate the checkpoint (intermediate parameters during training) for test time inference. On the experimental sides, the paper conduct experiments over standard dataset like CIFAR10 and stackoverflow and their methods improve the accuracy around 1% to 3%. It also provides some theory on convex optimization and demonstrate the advantage of averaging instead of using only the last iteration.

Method. The paper proposes to aggregate the checkpoint in a few ways: (1) average parameter (exponential averaging or uniform averaging) and (2) aggregate the output (majority vote or averaging). The DP guarantee follows directly as they are post-processing

Experiments. The paper conduct experiments over CIFAR10 and stackoverflow, in both centralized setting and federated setting. I am not a specialist in experiments but the improvement seems solid/consistent, but marginal at the same time.

Theory. The paper shows averaging is better than the last-iteration in the constrained convex optimization setting. It provides good insight. However, technically, their method just plug-in and play the bounds of [Shamir and Zhang'13], and they also fail to show a lower bound (this means it is not really a separation between averaging and plug-in and play, in contrast with [Shamir and Zhang'13]). Moreover, in the convex optimization setting, there are many DP gradient descent type algorithm that uses averaging, so it is not novel.



**Summary Of The Review:**

This a solid paper, but the idea/improvement seems to be marginal for a ICLR paper.

--------------------------------------------------------------------------------

Post rebuttal

I have read the rebuttal and my evaluation remains.

---

> ### Author Response · Authors · 2022-11-15
> **Thank you for your review!**
>
> Thank you for your time in reviewing our work! We’ve added a revised version of the paper, with changes in red. Here, we address some specific concerns:
>
> 1) **“aggregation of checkpoint seems to be fairly straightforward idea of post-processing”**
>
> Although our aggregation methods are simple, as we mention in our paper (e.g., Section 1, 2nd paragraph), in most prior works in DP ML and even in production settings, the common practice is to use only the final checkpoint for predictions. Our work is the first to comprehensively study various techniques aggregating intermediate checkpoints (at no additional privacy cost) and show that the benefits of which are significant across different settings and that such aggregations are compatible with any iterative DP algorithm.
>
> 2) **“The experiment improvement seems marginal”**
>
> It is important to note that the empirical improvements we demonstrate are for the SOTA training runs for the respective settings, so there is no additional privacy cost for the improvements. In terms of the magnitude of the improvements, for Stackoverflow next word prediction accuracy, checkpoints aggregation provide improvements of 0.43% for ε of 8.2 and 18.9 over DPFTRLM, which we believe is a significant gain. These improvements are comparable to those in prior SOTA works, e.g., [a] and [b] improved by ~0.57% for ε ~ 8.3 over DPSGDM. For CIFAR10, [a] improved by 0.6% and 2% for ε of 0.19 and 3.51, respectively, while we improve by 3.79% and 2.23% for ε of 1 and 8, respectively. Note that with higher ε (lower privacy), we expect gains to reduce in general.
>
> *[a] Amid et al., Public data-assisted mirror descent for private model training. ICML’22*
>
> *[b] Kairouz et al., Practical and private deep learning without sampling or shuffling, ICML’21*
>
> 3) **“theory only explains the insight at the convex optimization domain”**
>
> Our goal with Theorem 2.1 was not to prove any novel technical statement, but give theoretical support for the improvements in our experiments. While other methods give optimal last-iterate loss bounds, they usually require modifying the choice of e.g. step size or batch size to choices that are uncommon in practice. We have added citations to these methods for a complete comparison.

---

### Official Review · Reviewer_PDAE · 2022-10-25

**Confidence:** 4
**Correctness:** 4
**Technical Novelty And Significance:** 3
**Empirical Novelty And Significance:** 3
**Recommendation:** 6

**Clarity, Quality, Novelty And Reproducibility:**

The quality of this paper is solid and the clarity is good. My concern on the novelty of uncertainty quantification with DP, which has been studied in [1] (see Section 5). Therefore the argument in Section 3 "(Naive) uncertainty quantification, and its hurdles" are already known results. [1] studies DP-SGLD which includes DP-LD in this work's Section 3 as a sub-case. Though this work provides some theoretical analysis, using DP Bayesian approach to do uncertainty quantification is not new.

Also, using checkpoints is already explored as in EMA and DP-SGLD in [1][2] (e.g. Algorithm 2 in [1], the output line), whose computational advantage over multiple runs is clear so some novelty is comprised too.



[1] Bu et al. "Differentially Private Bayesian Neural Networks on Accuracy, Privacy and Reliability" (https://arxiv.org/pdf/2107.08461.pdf).
[2] Li et al. "On Connecting Stochastic Gradient MCMC and Differential Privacy" (https://arxiv.org/pdf/1712.09097.pdf)

**Strength And Weaknesses:**

Strength: This paper is clearly presented. The contribution is solid and the experiments are convincing and well-designed. The new method has no cost in terms of privacy guarantee, improves the accuracy, does no incur additional computation, and particularly the UPA trick is better than EMA in that no additional hyperparameter $\beta_t$ needs extra privacy budget to tune honestly.

Weaknesses: I have some concerns on the novelty (see details in next comment) and not comparing carefully to existing work. For example, although the work claims above Section 2.3.3 "we improve the SOTA baseline of De et al. (2022) from 70.6% to 75.51% at ε = 1 and from 77.6% to 80.8% at ε = 8". De et al. is not SOTA: CIFAR100 has achieved 83.0% at ε = 1 and 88.4% at ε = 8 without using any checkpointing like EMA, by "Scalable and Efficient Training of Large Convolutional Neural Networks with Differential Privacy". Testing on this setting is important.


**Summary Of The Paper:**

This paper studies re-using the intermediate checkpoints during DP training for two purposes: 1. improve the accuracy; 2. construct reasonable uncertainty quantification. The contribution is on the methodology and empirical results.

**Summary Of The Review:**

Overall I think this is a borderline paper and needs non-trivial revision on the related work discussion and experiments.

---

> ### Author Response · Authors · 2022-11-15
> **Thank you for your review!**
>
> Thank you for your time in reviewing our work! We’ve added a revised version of the paper, with changes in red. Here, we address some specific concerns:
>
> 1) **“not comparing carefully to existing work… CIFAR100 has achieved 83.0% at ε = 1 and 88.4% at ε = 8 without using any checkpointing like EMA”**
>
> Due to computational constraints, we use a 36M parameter WideResNet pre-trained on a downsampled ImageNet1k (from [a]). Moreover, for CIFAR100, since [a] observes that fine-tuning only the last layer provides better accuracy models as compared to fine-tuning all layers in high privacy (small ε) regimes, we show significant accuracy gains when fine-tuning only the last layer of the EMA checkpoint of the pretrained model, i.e., using checkpoint aggregation for pre-training as well. Note that [b] uses 303M parameter vision transformers pre-trained on ImageNet21k, and fine-tunes all the parameters to achieve 83.0% at ε = 1. We have added a brief explanation regarding this distinction in our paper (in Section 2.2.2, footnote-2 on page-5).
>
> *[a] De at al., Unlocking High-Accuracy Differentially Private Image Classification through Scale*
>
> *[b] Bu et al., Scalable and Efficient Training of Large Convolutional Neural Networks with Differential Privacy*
>
> 2) **“using DP Bayesian approach to do uncertainty quantification is not new”**
>
> To clarify, we are looking at Langevin diffusion in Theorem 3.1, which is slightly different from DP-SGLD considered in [1]. Furthermore, we do not claim that the connection between DP-SGD and Langevin diffusion is a novelty of our paper; the part of the paper where we discuss this connection is just meant to help a reader unfamiliar with Langevin diffusion/stochastic calculus understand what a Langevin diffusion is intuitively. Furthermore, we wish to clarify that we are not using a Bayesian approach like in [1]; instead we are proposing a much simpler approach that only uses information usually already present from training any model with DP-SGD, so BNNs are not needed.
>
> *[1] Bu et al. "Differentially Private Bayesian Neural Networks on Accuracy, Privacy and Reliability*

---

### Official Review · Reviewer_J6hu · 2022-11-04

**Confidence:** 3
**Correctness:** 3
**Technical Novelty And Significance:** 3
**Empirical Novelty And Significance:** 2
**Recommendation:** 5

**Clarity, Quality, Novelty And Reproducibility:**

Some authors' statements need more support . After Theorem 3.1, the authors claims that
> Per Harvey et al. (2019), in general this log(n) factor cannot be removed by a better analysis of the last iterate of DP-SGD as instantiated in Theorem 2.1.

However, Theorem 2.1. does not provide any technical lower bound that about why the log(n) factor "cannot" be removed. The reference Harvey et al. (2019) also does not provide any privacy analysis. Therefore it is difficult to understand how this statement is supported. The authors may want to add more detailed explanations. Finally, it is not clear what "in general" means. The authors may want to clarify the conditions for this claim, and add more reference for other settings where the log(n) factor could be removed, such as [a] [b] and [c]. Otherwise, the current claim is too strong and potentially misleading.

[a] Jain, P., Nagaraj, D., & Netrapalli, P. (2019, June). Making the last iterate of sgd information theoretically optimal. In Conference on Learning Theory (pp. 1752-1755). PMLR.
[b] Feldman, V., Koren, T., & Talwar, K. (2020, June). Private stochastic convex optimization: optimal rates in linear time. In Proceedings of the 52nd Annual ACM SIGACT Symposium on Theory of Computing (pp. 439-449).
[b] Chourasia, R., Ye, J., & Shokri, R. (2021). Differential privacy dynamics of langevin diffusion and noisy gradient descent. Advances in Neural Information Processing Systems, 34, 14771-14781.

**Strength And Weaknesses:**

Strengths:
- Averaging intermediate models is a promising direction for improving privacy-accuracy trade-off for differentially private learning.
- Utilizing checkpoints for uncertainty estimates is an attractive methodology that may enjoy lower computation cost.

Weaknesses:
- In terms of empirical result, it's not really clear what averaging method one should use and why. In the paper, the various aggregation methods (including one method from existing work De et al.) perform similarly for most tasks. For a few tasks where the differences between methods are more significant, the winning method is also not consistent. The authors may want to clarify why their methods are better than prior works for certain tasks but not others. How does this depend on the data distribution or learning tasks?
- In terms of theoretical results, to my understanding, the paper's main contribution is Theorem 3.1. which shows improved uncertainty (variance) estimate as the burning time and intervals between checkpoints increase. However, I do not see the effect of the number of checkpoint models in this theorem. It's not clear whether we should use more checkpoints for variance estimation and why.
- In terms of the motivation of using checkpoints for uncertainty estimate, the authors emphasize computation efficiency of re-using checkpoints when compared to training multiple independent last-iterate models. However, the total number of training epochs increase in order to produce good quality checkpoints (with reasonable interval between them). These computations are not parallelizable. I think the authors need to support this motivation with more evidence, e.g., how much is the additional computation cost for longer training? Is it negligible when compared to multiple independent runs of the training algorithm (which could be done in parallel)?

**Summary Of The Paper:**

The paper investigates whether aggregating intermediate checkpoints of DP-SGD algorithms enable better privacy-accuracy trade-off and uncertainty estimate (when compared to those of last-iterate model). They provide empirical evidence that a variety of check point aggregation methods enables improved test accuracy for centralized and federated learning tasks. They also show that for strongly convex smooth loss function, check-point aggregation enables accurate variance estimate for bounded statistics of last-iterate model. This estimation error decays exponentially with the burn-in time and the minimum interval between any two sampled checkpoints.

**Summary Of The Review:**

The paper shows interesting potential of using intermediate model aggregation to improve privacy-accuracy trade-off and uncertainty estimate of differentially private learning. However, at current state, the paper does not offer clear investigation of which aggregation method works best in different setting and why. And the paper's Theorem 3.1, motivation for aggregating checkpoints, and some of the claims need more support and explanations. Therefore, my assessment is borderline.


======
Post authors' response

My main concern is still that the paper still needs better (more consistent) explanations for the gain in privacy-utility trade-off or uncertainty estimate by utilising intermediate checkpoints.

=====
Response to author's followup comments:
> The increase in training process length is unavoidable for a worst-case analysis; even for a simple loss such as a one-dimensional quadratic loss, obtaining k near-independent samples from a distribution close to the distribution of DP-SGD’s outputs using o(k) times the number of gradient accesses as a run of DP-SGD (i.e., o(k) times the privacy budget) would violate privacy lower bounds.

I appreciate the authors' lower bound argument. But at the current form, it needs more explanation, perhaps a formal statement and some references, to be interpretable and convincing.

Secondly, "obtaining k near-independent samples from a distribution" is not a necessary condition for good uncertainty estimate. There are many uncertainty estimation methods, such as MC-dropout, that rely on correlated samples. The idea of using intermediate checkpoints intrinsically may involve analyzing how to use correlated samples and their benefits. Reducing the analysis to nearly-independent runs of MCMC (via large checkpoint interval) lack the perspective for explaining the advantage of performing checkpoint aggregation compared to independent runs of the training (fine-tuning) process.

> We disagree with this performance analysis of UPA (i.e., uniform tail average). First note from Tables 1 to 3 and Figures 1 and 2 that UPA always performs better than the last checkpoint by large margins. Furthermore, note from Tables 1 and 2 that for the Stackoverflow dataset, UPA always performs better than all other aggregation methods.

In Table 1 and Figure 1 for the performance of UPA: For CIFAR10, under epsilon = 8, the performance improvement of UPA is 2.21%, which is only 0.33% larger than the standard deviation of the baseline (1.46%) plus the standard deviation of UPA (0.52%). Therefore, this test accuracy improvement almost falls in the range of standard deviation, which is marginal. For stackoverflow, in Table 1, the test accuracy improvement of UPA is below 0.5%, which is marginal.

The test accuracy gain of UPA in Table 2-3 and Figure 2 is indeed more significant. However, the experiments for Table 2 and 3 and Figure 2 are under periodic distribution shift or data-dependent checkpoints aggregations, which the current Theorem 2.1 does not apply to. To explain the benefits in Table 2 and 3 and Figure 2, the authors would need new analysis.

---

> ### Author Response · Authors · 2022-11-15
> **Thank you for your review!**
>
> Thank you for your time in reviewing our work! We’ve added a revised version of the paper, with changes in red. Here, we address some specific concerns:
>
> 1) **"not really clear what averaging method one should use and why… How does this depend on the data distribution or learning tasks?"**:
>
> In general, we do not expect a single aggregation method to work best for all settings due to differences in types of tasks (image, text, etc.), data distributions (e.g., diurnal, uniform, etc.), and training regimes (model architecture, type of DP training method, etc.). However, note that due to the post-processing property of DP, one can try various aggregation methods to find the best one without any additional privacy cost, as intermediate checkpoints already satisfy the DP guarantee of the training method. Finally, we would like to point out that for Stackoverflow, uniform past averaging (UPA) always works the best (Figures 1, 3 and Tables 1, 3) and provides significant improvements over prior SOTA; in case of CIFAR10 output predictions averaging (OPA) almost always performs the best (albeit with smaller improvements).
>
> 2) **“not clear whether we should use more checkpoints for variance estimation and why”**
>
> The theorem was mainly meant to give intuition for why the covariance between checkpoints might be small in practice, and not to inform the choice of how many checkpoints to use in variance estimation. However, it is fairly straightforward to take the theorem and make it a statement about the number of checkpoints that minimizes a bound on the bias-variance tradeoff of the sample variance estimator. We have added this discussion to the submission in the appendix (see Appendix A.1).
> The high-level idea: We can show the variance of this estimator is at most the variance of iid samples (which is at most 1/k) plus our bias bound (which is exponentially decaying in something like b/k - a, where b and a are constants wrt k). So the bias + variance of this estimator is at most something like 1/k + exp(b/k - a), and we can find the k minimizing this term.
> “how much is the additional computation cost for longer training…compared to multiple independent runs”: We note that our theoretical result on uncertainty estimation is just meant to give intuition for why checkpoints in our empirical evaluation may have low covariance. Empirically, the computation and privacy cost of running multiple runs is significantly higher than running a single run, as in our experiments, we do not change the number of rounds of a training run.
>
> 3) **“Per Harvey et al. (2019), in general this log(n) factor cannot be removed by a better analysis of the last iterate of DP-SGD as instantiated in Theorem 2.1.”**
>
> Our goal with Theorem 2.1 was not to prove any novel technical statement, but give theoretical support for the improvements in our experiments. However, we agree that the phrasing of this statement is unclear/strong as written and will revise it. We will change it to clarify that (i) we do not know how to remove the last-iterate log factor for vanilla DP-SGD (ii) this statement is specifically for SGD with e.g., fixed batch size and fixed or polynomially decaying step sizes, which are common in practice and (iii) is a statement about high-probability bounds. We have included references to the works removing the log factor, although we feel averaging is preferable since it is a simple post-processing (unlike [a] and [b] which require modifying the learning rate schedule/batch size; [c] requires constant condition number) that we demonstrated works well empirically.

---

> > ### Comment · Reviewer_J6hu · 2022-12-09
> > **Thank you for the clarifications**
> >
> > The authors clarified my concerns about the computation efficiency of using checkpoints for uncertainty estimation and provided more explanations for the log(n) factor in the privacy utility trade-off.
> >
> > However, the paper's theoretical results and empirical evaluations still seem disconnected, even after the author's response. For example, as the authors stated in the rebuttal, a longer training process is unnecessary to generate good quality checkpoints for uncertainty estimates in empirical evaluations. However, for the theoretical analysis, Theorem 3.1 still requires a significantly longer training process to reach good checkpoints for uncertainty estimates. As another example, the theoretical results for the gain of privacy utility trade-off in Theorem 2.1 assumes a uniform tail average. In contrast, in experiments, this uniform average method only sometimes gives better accuracy than last-iterate, and generally performs worse than other averaging methods.
> >
> > In conclusion, I feel that the paper still needs better (more consistent) explanations for the gain in privacy-utility trade-off or uncertainty estimate by utilising intermediate checkpoints.

---

> > > ### Author Response · Authors · 2022-12-11
> > > **Thanks for the follow up comments!**
> > >
> > > Thanks for the follow up comments! Below are our responses to the specific concerns.
> > >
> > > >"As the authors stated in the rebuttal, a longer training process is unnecessary to generate good quality checkpoints for uncertainty estimates in empirical evaluations. However, Theorem 3.1 still requires a significantly longer training process to reach good checkpoints for uncertainty estimates."
> > >
> > > **Response**: The increase in training process length is unavoidable for a worst-case analysis; even for a simple loss such as a one-dimensional quadratic loss, obtaining k near-independent samples from a distribution close to the distribution of DP-SGD’s outputs using o(k) times the number of gradient accesses as a run of DP-SGD (i.e., o(k) times the privacy budget) would violate privacy lower bounds. We would like to reiterate that Theorem 3.1 is not advocating using a longer training process for uncertainty estimations, but instead meant to give intuition for why checkpoints may not be heavily correlated, as in our empirical study. Many theoretical results on convex optimization in the literature are somewhat disconnected from the corresponding empirical results, and we feel our paper is not out of line in this regard.
> > >
> > > >“... theoretical results in Theorem 2.1 assumes a uniform tail average. In contrast, in experiments, this uniform average method only sometimes gives better accuracy than last-iterate, and generally performs worse than other averaging methods.”
> > >
> > > **Response**: We disagree with this performance analysis of UPA (i.e., uniform tail average). First note from Tables 1 to 3 and Figures 1 and 2 that UPA always performs better than the last checkpoint by large margins. Furthermore, note from Tables 1 and 2 that for the Stackoverflow dataset, UPA always performs better than all other aggregation methods. Finally, as we clarified before, due to the post-processing property of DP, one can find the best aggregation method using DP checkpoints at no additional privacy cost for a given setting.

---

> > > > ### Author Response · Authors · 2022-12-13
> > > > **Thanks for the follow up comments!**
> > > >
> > > > Thanks for the follow up comments! Below are our responses to the specific concerns.
> > > >
> > > > > Concerns about the uncertainty estimation ("I appreciate the authors' lower bound argument...")
> > > >
> > > > We agree that obtaining k near-independent samples is not necessary for a reasonable variance estimate (indeed, even Theorem 3.1 can be used to show sufficient theoretical conditions for having moderate rather than near-zero correlation between checkpoints); our response is meant to explain why an analysis that allows for arbitrarily small correlation between checkpoints such as ours would require an increase in the training time, not to argue this is necessary in general. As the reviewer mentions, it seems somewhat likely that the bias of the sample variance of checkpoints must depend on the correlation between them, which maybe is undesirable in light of other methods. But we still feel it is important to understand since methods for uncertainty estimation like MC Dropout and the use of BNNs are model-specific, whereas using the sample variance of loss at the checkpoints is not.
> > > >
> > > > And we feel the important takeaway from Theorem 3.1 should not be that one can reduce to nearly independent runs, but rather that since DP-SGD is a Markov chain approaching a stationary distribution (from any point initialization), this implies the dependence between two checkpoints is decreasing to 0 over time. While our theoretical analysis looks at this in the case where we allow enough time for the Markov chain to converge to near its stationary distribution, one should still expect some amount of convergence towards a stationary distribution and similar decreases in dependence even with smaller times between checkpoints. This is even reflected in part of our analysis, where we show that starting from two point initializations (with infinite divergence), the divergence becomes finite and exponentially decreasing relatively quickly.
> > > >
> > > > > Concerns about the accuracy gains ("In Table 1 and Figure 1 for the performance of UPA... ")
> > > >
> > > > - *General statement about the practical utility of our proposals*: First, we would like to point out that the gains due to our proposals are free of privacy cost and do not need any changes to the current DP training algorithms. In other words, unlike many DP improvements, our proposals are practically easy to adapt to and improve privacy-utility trade-offs.
> > > >
> > > > - *On the accuracy gains due to our proposals*: Note that, for Stackoverflow with 10k total output classes, we believe that 0.5% (absolute) accuracy gain over previous SOTA DP training algorithm, i.e., DPFTRLM (Kairouz et al., 2021), is a significant improvement. Please check [a], which proposes a state-of-the-art DP algorithm, with ~0.5% accuracy gains (Fig 2b in [a]) for Stackoverflow and 0.7% accuracy gain for CIFAR10 (Table 2 in [a]). Finally, note that at higher epsilons, we expect diminishing gains due to any novel DP learning method, which we also observe for CIFAR10 at an epsilon of 8. However, we strongly believe this does not discredit the utility of our proposals, as the gains at lower epsilons are significant.
> > > >
> > > > - *About Theorem 2.1*: As clarified before and revised in the new draft, this theorem does not aim to provide any novel technical statement, but gives theoretical intuition for the improvements in our experiments. Theorem 2.1 intuitively shows that averaging mitigates the variance of gradients across rounds when the examples in each round are sampled i.i.d. While this does not hold in the distribution shift setting, we feel it is intuitive that averaging can only help mitigate the variance in gradients further in this setting, even without theoretical justification. This is demonstrated by our experiments, and as the reviewer acknowledges, the gains are “indeed more significant” in this setting.
> > > >
> > > > [a] Amid et al., Public data-assisted mirror descent for private model training. ICML’22

---

### Decision · Program_Chairs · 2023-01-20

**Decision:**

Reject

**Justification For Why Not Higher Score:**

Two of the reviewers (which are expert reviewers) have recommended rejecting the paper.

**Justification For Why Not Lower Score:**

-

**Metareview: Summary, Strengths And Weaknesses:**

The paper studies several methods that aggregate intermediate checkpoints (during training) to improve the utility of DP training. The authors propose two  classes of aggregation methods based on aggregating the parameters of checkpoints, or their outputs. Moreover, the authors consider the problem of quantifying the uncertainty that DP noise adds to DP ML training, and show that the sample variance from last few checkpoints provides a good approximation of the variance of the final model of a DP run. The proposed methods are supported by theoretical statements and empirical evaluations.

All the authors found the ideas of the paper to be interesting. They also thought hat the paper was well written. I have discussed this papers with the reviewers, and some of their major concerns still remain. In particular (please see the updated reviews for full details), they were not convinced by (i) the statements/explanations provided in the paper on the gain in privacy-utility trade-off or uncertainty estimate by utilizing intermediate checkpoints; (ii) the experimental results need clear revisions/improvements. There were also some other concerns about the theoretical results, etc.

All in all, I this the paper introduces a novel idea of using intermediate model aggregation to improve privacy-accuracy trade-off. Once the reviewers' concerns are properly addressed, the paper will become an interesting contribution to the literature of private learning.

**Summary Of Ac-Reviewer Meeting:**

This was a borderline paper. I have been discussing the paper with some of the reviewers (via email -- no meeting was necessary in this case. I also know the expert reviewer of this paper very well, and we chat quite frequently). The reviewers have major concerns that were not addressed by the authors.